# Robust and prototypical immune responses toward COVID-19 vaccine in First Nations peoples are impacted by comorbidities

Wuji Zhang [1], Lukasz Kedzierski [1,2], Brendon Y. Chua[1], Mark Mayo[3], Claire Lonzi[3], Vanessa Rigas[3], Bianca F. Middleton[3], Hayley A. McQuilten [1], Louise C. Rowntree[1], Lilith F. Allen[1], Ruth A. Purcell [1], Hyon-Xhi Tan [1], Jan Petersen[4], Priyanka Chaurasia [4], Francesca Mordant [1], Mikhail V. Pogorelyy[5], Anastasia A. Minervina[5], Jeremy Chase Crawford [5], Griffith B. Perkins [6,7], Eva Zhang[8,9], Stephanie Gras [4,10], E. Bridie Clemens[1], Jennifer A. Juno [1], Jennifer Audsley[11], David S. Khoury [12], Natasha E. Holmes[13], Irani Thevarajan[11,14], Kanta Subbarao[1,15], Florian Krammer [16], Allen C. Cheng[17,18], Miles P. Davenport [12], Branka Grubor-Bauk[7], P. Toby Coates[6,7], Britt Christensen[7,19], Paul G. Thomas [5], Adam K. Wheatley [1], Stephen J. Kent [1,20,21], Jamie Rossjohn [4,22], Amy W. Chung [1], John Boffa[23], Adrian Miller[24], Sarah Lynar[3,25], Jane Nelson[3], Thi H. O. Nguyen [1,27] ✉, Jane Davies [3,27] ✉ & Katherine Kedzierska [1,26,27] ✉

High-risk groups, including Indigenous people, are at risk of severe COVID-19. Here we found that Australian First Nations peoples elicit effective immune responses to COVID-19 BNT162b2 vaccination, including neutralizing antibodies, receptor-binding domain (RBD) antibodies, SARS-CoV-2 spike-specific B cells, and CD4$^+$ and CD8$^+$ T cells. In First Nations participants, RBD IgG antibody titers were correlated with body mass index and negatively correlated with age. Reduced RBD antibodies, spike-specific B cells and follicular helper T cells were found in vaccinated participants with chronic conditions (diabetes, renal disease) and were strongly associated with altered glycosylation of IgG and increased interleukin-18 levels in the plasma. These immune perturbations were also found in non-Indigenous people with comorbidities, indicating that they were related to comorbidities rather than ethnicity. However, our study is of a great importance to First Nations peoples who have disproportionate rates of chronic comorbidities and provides evidence of robust immune responses after COVID-19 vaccination in Indigenous people.

Higher morbidity and mortality rates from coronavirus disease 2019 (COVID-19) are disproportionately observed in high-risk groups, including Indigenous people[1,2]. While epidemiological reports suggest that Indigenous people are more susceptible to severe acute respiratory syndrome coronavirus 2 (SARS-CoV-2) infection[2,3], immunological data on immune responses to SARS-CoV-2 infection and vaccination in Indigenous populations are lacking. There are approximately 476 million Indigenous people globally, including more than 798,300

Aboriginal and Torres Strait Islanders (respectfully referred to as Australian First Nations (FN) peoples)[1]. Indigenous populations in Brazil and the USA had higher COVID-19 cases and a higher case fatality ratio[4]. Native Americans and Alaskan Natives were three times more likely to be hospitalized and more than twice as likely to die from COVID-19 than non-Indigenous (NI) populations, with recent data showing a decline in life expectancy of 6.5 years since 2019 as a result of COVID-19 (ref. 3). Australian FN peoples, Native Americans and Alaskan Natives have disproportionate rates of diabetes and chronic respiratory and renal disease, with death rates from these chronic conditions before COVID-19 higher than in NI populations[5]. COVID-19 epidemiological data from Australian FN peoples is lacking, stemming from low SARS-CoV-2 infection rates during the early 'zero COVID-19 policy' in Australia, followed by high immunization rates and access to treatments.

Indigenous populations experience higher rates of tuberculosis[6], sepsis[7] and viral infections, including influenza[8]. Hospitalization, intensive care unit admission and morbidity rates were increased in Australian FN peoples compared to NI Australians during the 2009 H1N1 influenza pandemic[9]. Higher H1N1 influenza rates were also observed in native Brazilians[10], Alaskan Natives and Native Americans[11], New Zealand Maori and Pacific Islanders[12]. Socio-economic factors[2] can contribute to increased infection rates in Indigenous communities. However, it is unknown whether the higher morbidity and mortality and prolonged hospitalization may be also explained by perturbed immunity toward viral pathogens.

In healthy NI adults, the Pfizer BioNTech BNT162b2 vaccine[13] can induce robust antibody and T cell responses toward the ancestral SARS-CoV-2 strain[14-16], with T cells providing conserved responses against variants of concern (VOCs)[17]. To date, studies assessing COVID-19 vaccine immunogenicity in Indigenous people are missing. Such knowledge is needed to inform vaccine regimens and immunotherapies to best protect Indigenous populations from severe COVID-19.

We recruited FN and NI Australian people vaccinated with BNT162b2 in 2021–2022 and assessed their immunity before and after vaccination at six time points. We performed antibody analyses toward the ancestral SARS-CoV-2 strain and VOCs, and assessed B cell and T cell activation ex vivo using spike-specific probes, peptide-HLA class I and class II tetramers, and activation-induced marker (AIM) and intracellular cytokine secretion (ICS) assays. We found that Australian FN peoples mounted effective immune responses to BNT162b2. However, receptor-binding domain (RBD) antibodies, spike-specific B cells and follicular helper T ($T_{FH}$) cells were reduced in Australian FN peoples with comorbidities (CMs), linked to elevated levels of agalactosylated bulk IgG and interleukin-18 (IL-18). Reduced SARS-CoV-2 antibody and B cell responses, with altered glycosylation patterns and increased IL-18 were also observed in NI people with CMs. Our study is of importance to FN peoples who have disproportionate rates of CMs, including diabetes and renal disease, and provides in-depth data on immune responses after COVID-19 vaccination in Indigenous people.

## Results

### RBD antibodies increase after vaccination in FN peoples

A total of 97 SARS-CoV-2-unexposed, seronegative participants who received the BNT162b2 vaccine (hereafter mRNA vaccine) (58 Australian FN peoples and 39 NI individuals; Fig. 1a) were recruited into the COVAC cohort through the Menzies School of Health Research in Darwin, Northern Territory, Australia. Median age was 44 years (range 19–79 years) in the Australian FN peoples cohort and 44 years (range 23–64 years) in the NI cohort; 47% of participants in the Australian FN peoples cohort were female, whereas 74% of NI participants were female (Fig. 1b). Sampling was performed before dose 1 (V1), at day 6–28 after dose 1 (V1a), before dose 2 (V2), day 28 after dose 2 (V3), month 6 after dose 2 (V4) and day 28 after dose 3 (V5) (Fig. 1a and Supplementary Table 1).

We analyzed IgG antibody responses directed at RBD[18,19] corresponding to the ancestral (vaccine) strain across all time points and Delta/Omicron at V1, V3 and V5 (Fig. 1c). RBD IgG antibody levels significantly increased from V1 after the first vaccine dose in both groups and peaked at V3, before declining at V4 to levels observed at V2 (Fig. 1d). At the peak of the response (V3), the median RBD IgG titer was slightly but significantly lower in Australian FN peoples ($\log_{10}$ titer = 3.9) than in NI participants ($\log_{10}$ titer = 4.2), as a few Australian FN peoples were below the seropositive cutoff (4 of 46, 8.7%) compared to none in NI participants (Fig. 1d). However, after the decline in antibodies observed at V4, dose 3 vaccination in a subset of Australian FN peoples induced significantly higher levels of antibodies at V5, with a higher median RBD IgG titer ($\log_{10}$ = 4.2) than the initial peak V3 responses ($\log_{10}$ = 3.9) (Fig. 1e).

Seroconversion levels (fourfold or greater increase) were similar in FN and NI peoples at 92% and 100% at V3, and 81% and 89% at V4 (Fig. 1f). Dose 2-induced RBD IgG titers in FN peoples increased toward the Delta variant, similar to but slightly lower than the NI group (Fig. 1g); both groups had low levels of antibodies against Omicron after dose 2 (Fig. 1g). The median IgG titer against Delta and Omicron RBD increased after dose 3 (Fig. 1h), correlating with IgG levels against the ancestral strain in both groups (Fig. 1i). Neutralizing antibodies increased after two doses to comparable levels in Australian FN and NI peoples (Fig. 1j). Neutralizing antibodies were detected at V3 and V4 for the ancestral and Delta strains (Fig. 1j), indicating high cross-reactive capacity of neutralizing antibodies toward the Delta variant, as reported elsewhere[18]. Overall, both FN and NI cohorts had robust RBD and neutralizing antibody responses.

### Increased bulk IgG G0 is associated with lower RBD IgG

Despite the same median age for both cohorts, we found an inverse correlation between age and RBD IgG titers in FN participants at V3, while RBD IgG titers correlated with their body mass index (BMI) (Fig. 2a). Such correlations were not observed in NI peoples. While female participants represented 74% in the NI cohort compared to 47% in FN peoples, two-way analysis of variance (ANOVA) with Šidák's multiple comparisons revealed higher ($P = 0.0362$) antibody titers in NI than FN female participants (Fig. 2a). Australian FN peoples with CMs including renal disease and diabetes were generally older than those without CMs and had lower IgG titers at V3 (Fig. 2b,c), indicating that CMs rather than the age per se might be a determining factor for reduced anti-RBD IgG levels in Australian FN peoples. However, most Australian FN peoples with or without CMs were still above the seropositive cutoff line (Fig. 2c). CMs were not reported in NI participants, while 36% (21 out of 58) of Australian FN peoples reported chronic diabetes or renal disease (or both) (Supplementary Table 1).

Differences in IgG glycosylation were reported across different geographical regions such as North America and Africa[20] and between seroconverters and non-seroconverters[21]. Agalactosylated IgG antibodies are regarded as pro-inflammatory[22] and are associated with lower antibody titers during influenza vaccination[23,24]. Total abundance of IgG G0 (no galactose) at V1 was greater in FN Australians than in NI participants (Fig. 2d). This was reflected in pronounced increases in the abundance of G0f (core fucose/no galactose) in Australian FN peoples at V1 (1.6-fold) and V3 (1.6-fold) compared to NI participants (Fig. 2d). Total IgG G2 (two galactose units) abundance was increased in NI compared to FN peoples but not for total IgG G1 (Fig. 2d), whereas median IgG G0 glycosylation was stable after vaccination (Fig. 2e).

Higher V1 IgG G0 glycosylation correlated with lower V3 RBD-specific IgG titers for Australian FN peoples and NI participants (Fig. 2f). FN peoples with CMs had higher IgG G0 abundance, and lower IgG fucose and sialylation G2S1f abundance, compared to FN peoples without CMs (Fig. 2g). G0 abundance correlated with age for both FN

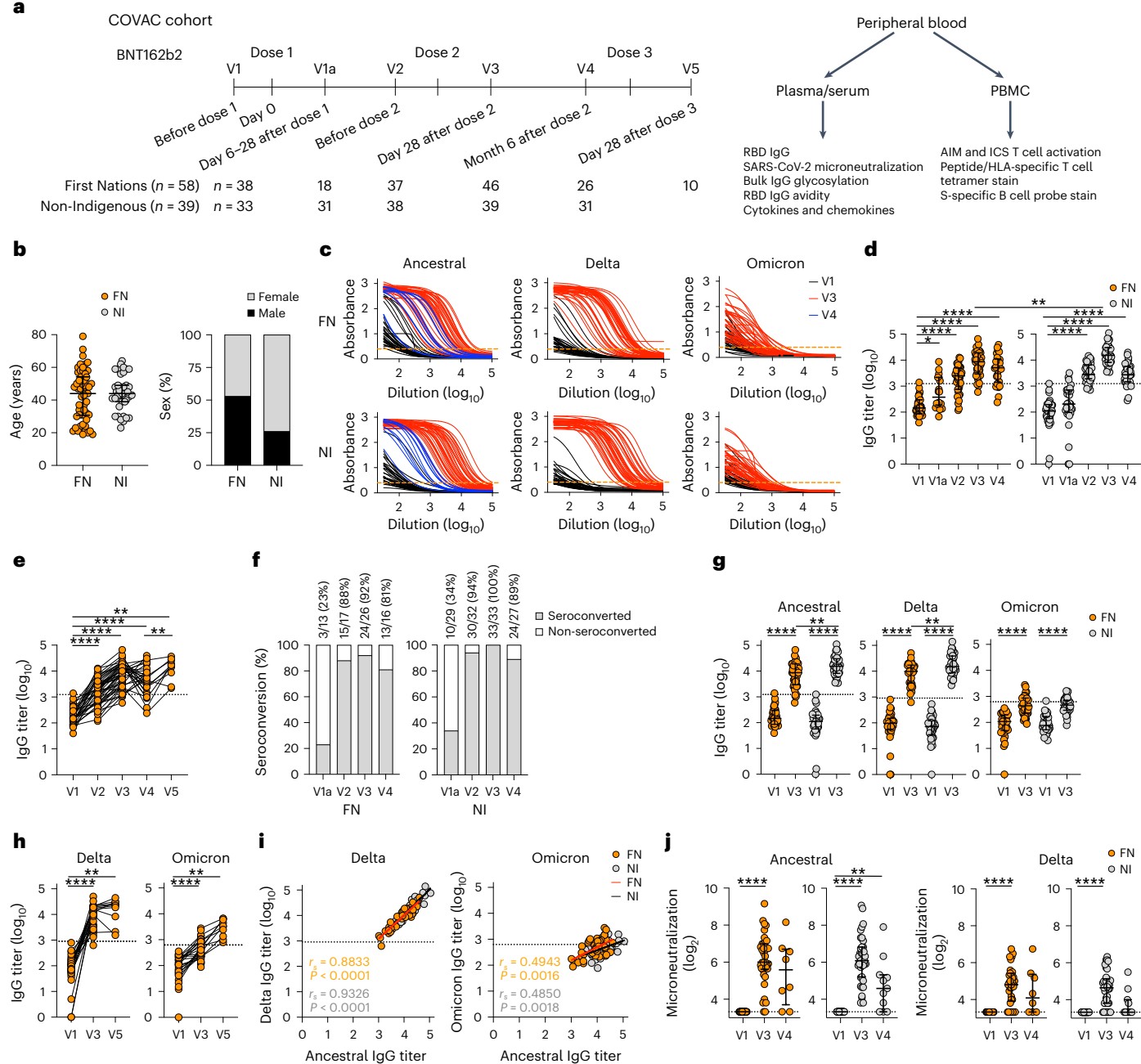

**Fig. 1 | Robust IgG responses toward ancestral and Delta RBD in FN and NI cohorts after mRNA vaccine vaccination. a**, Study and experimental design of the COVAC cohort in which blood samples were collected from 58 FN and 39 NI individuals. S, spike protein. **b**, Age and sex distribution of COVAC participants as in **a** ($n_{FN}$ = 58, $n_{NI}$ = 39). **c**, ELISA of RBD IgG showing the titration curves of absorbance (450 nm). The orange dashed lines indicate the endpoint titer cutoffs. **d,e**, ELISA showing ancestral RBD IgG at V1, V1a, V2, V3 and V4 (FN: $n_{V1}$ = 38, $n_{V1a}$ = 18, $n_{V2}$ = 37, $n_{V3}$ = 46, $n_{V4}$ = 26, V1 versus V1a $P$ = 0.0171; NI: $n_{V1}$ = 33, $n_{V1a,V4}$ = 31, $n_{V2}$ = 38, $n_{V3}$ = 39, V3 FN versus NI $P$ = 0.0013) (**d**) and V5 ($n_{V5}$ = 10; V1 versus V5, $P$ = 0.0078; V4 versus V5, $P$ = 0.0078) (**e**). **f**, Seroconversion rate (%) of ancestral RBD IgG (FN: $n_{V1a}$ = 13, $n_{V2}$ = 17, $n_{V3}$ = 26, $n_{V4}$ = 16; NI: $n_{V1a}$ = 29, $n_{V2}$ = 32, $n_{V3}$ = 33, $n_{V4}$ = 27) defined as at least a fourfold increase in $\log_{10}$-transformed titer

compared to V1. **g,h**, ELISA showing the Delta and Omicron RBD IgG at V1, V3 (FN: $n_{V1}$ = 32, $n_{V3}$ = 37 (Delta), $n_{V3}$ = 38 (Omicron); NI: $n_{V1}$ = 33, $n_{V3}$ = 39; V3 FN versus NI, $P$ = 0.0013 (ancestral) and $P$ = 0.0056 (Delta)) (**g**) and V5 ($n_{V5}$ = 10; V1 versus V5 $P$ = 0.0078 (Delta, Omicron)) (**h**). **i**, Spearman correlations between V3 ancestral RBD IgG and V3 Delta or Omicron RBD IgG (FN: $n_{Delta}$ = 38 and $n_{Omicron}$ = 39; NI, $n$ = 39). **j**, Microneutralization assay showing neutralizing antibodies against ancestral and Delta SARS-CoV-2 at V1, V3 and V4 (FN: $n_{V1}$ = 31, $n_{V3}$ = 38, $n_{V4}$ = 8; NI: $n_{V1}$ = 33, $n_{V3}$ = 39, $n_{V4}$ = 11, NI V1 versus V4, $P$ = 0.0078 (ancestral)). The bars indicate the median with the interquartile range (IQR). The black horizontal dotted lines indicate seropositivity. Statistical significance was determined with a two-sided Wilcoxon test in each cohort or a two-sided Mann–Whitney $U$-test between FN and NI cohorts. *$P$ < 0.05, **$P$ < 0.01, ***$P$ < 0.001, ****$P$ < 0.0001.

and NI peoples (Fig. 2h). Using a urea dissociation assay[25,26], we found no differences in the avidity index for RBD-specific IgG antibodies in FN and NI peoples with or without CMs at V3 (Fig. 2i), while RBD-specific IgG titers were strongly correlated with the avidity index (Fig. 2j).

Higher plasma concentrations of the soluble mediators monocyte chemoattractant protein-1 (MCP-1), IL-8 and IL-18, and lower concentrations of IL-1β, interferon-α2 (IFNα2), IFNγ, tumor necrosis factor (TNF), IL-10, IL-12p70, IL-17A, IL-23 and IL-33 were detected in

Australian FN participants (with and without CMs) compared to NI participants at V1 (Fig. 2k and Extended Data Fig. 1a). IL-1β, IFNα2, IL-12p70, IL-23 and IL-33 were further reduced in Australian FN peoples with CMs compared to those without (Extended Data Fig. 1b). IL-18, an IFNγ-inducing pro-inflammatory cytokine, was higher in Australian FN individuals with CMs compared to those without (Fig. 2l, 283 and 508 pg ml$^{-1}$, respectively) and was correlated with G0 abundance (Fig. 2m). Consequently, IL-18 levels were conversely correlated with RBD IgG titers at V3 (Fig. 2m).

Frequency of spike-probe-specific IgD$^-$ memory B cells increased from V1 to V4 in the Australian FN and NI cohorts (Fig. 2n). The frequency of spike-specific IgD$^-$ B cells at V4 was lower in FN peoples than in NI peoples (Fig. 2n) due to reduced frequencies of spike-specific B cells in Australian FN peoples with CMs (Fig. 2o). The frequency of spike-specific IgD$^-$ B cells did not correlate with RBD IgG titers at V3 or V4 (Fig. 2p), confirming reports of increased SARS-CoV-2-specific B cells over time after infection[27]. As such, higher total IgG G0 glycosylation profiles at baseline in FN peoples with CMs were inversely correlated with SARS-CoV-2-specific IgG antibody levels after mRNA vaccination.

## RBD IgG is reduced and IgG G0 is increased in NI peoples with CMs

To test whether correlation between reduced SARS-CoV-2 antibody responses after COVID-19 vaccination and increased G0 abundance at baseline related to ethnicity or CMs, we obtained samples from an additional 69 NI individuals, including 38 individuals with diabetes or renal disease (or both) (16 dissection of influenza-specific immunity (DISI) participants with V1 samples and 22 participants with V1, V3 or V4 samples) and 31 individuals with inflammatory bowel disease (IBD) (V1, V3 and V4 samples). DISI, NI with diabetes or renal disease (or both) and participants with IBD had a median age of 54, 58 and 30 years and 50% (8 of 16), 36.3% (8 of 22) and 51.6% (16 of 31), respectively were female (Supplementary Table 2). NI peoples with CMs had decreased RBD antibody responses (Fig. 3a) and elevated G0 abundance (Fig. 3b) compared to FN and NI peoples without CMs, although no association was observed with the V3 RBD IgG titer (Fig. 3c). IL-18 was increased at V1 in NI individuals with CMs compared to those without (Fig. 3d) and was inversely correlated with RBD IgG titers in individuals with IBD (Fig. 3e). We found reduced frequencies of spike-specific B cells in NI participants with IBD (Fig. 3f). Multiple linear regression confirmed that CMs, G0 abundance and IL-18 were predictors of reduced antibody responses after adjusting for age, sex and BMI. These data indicated an inverse correlation between antibody responses after SARS-CoV-2 vaccination and G0 abundance of IgG antibodies at V1 in FN and NI individuals, which was related to CMs rather than ethnicity.

## CD134$^+$CD137$^+$ T$_{FH}$ cell frequency is decreased in people with CMs

Global SARS-CoV-2-specific CD4$^+$ and CD8$^+$ T cell responses at V1, V3, V4 and V5 were assessed with AIM and ICS (Extended Data Fig. 2a,b). Frequency of spike-specific CD134$^+$CD137$^+$CD4$^+$ and CD69$^+$CD137$^+$CD8$^+$ T cells increased between V1 and V3 in the FN and NI cohorts (Fig. 4a,b). Frequency of CD69$^+$CD137$^+$CD8$^+$ T cells at V3 was lower in FN peoples than in NI peoples (median 0.04% and 0.12%, respectively) (Fig. 4b). In FN peoples, median frequency of CD134$^+$CD137$^+$CD4$^+$ and CD69$^+$CD137$^+$CD8$^+$ T cells decreased at V4, although it was higher than V1 levels and stable at V5 (Fig. 4c).

Functional TNF$^+$IFNγ$^+$ ICS analysis (Fig. 4d) showed increases in spike-specific CD4$^+$ T cell responses at V3 compared to V1 (Fig. 4e), with FN peoples having a slightly higher frequency of TNF$^+$IFNγ$^+$CD4$^+$ T cells compared to NI participants at V4 (Fig. 4e). The frequency of TNF$^+$IFNγ$^+$CD8$^+$ T cells was higher in Australian FN participants at V3 than V1 and comparable between the FN and NI cohorts (Fig. 4e,f). The spike-specific T cell responses detected with AIM and ICS assays were comparable in Australian FN and NI individuals at V3, irrespective of CMs (Fig. 4g,h).

Circulating ICOS$^+$PD-1$^+$CD4$^+$ T$_{FH}$ cells emerge during SARS-CoV-2 infection[25,28] and vaccination[15] and are associated with antibody responses (Fig. 4i). Spike-specific CD134$^+$CD137$^+$ responses for T$_{FH}$ cells, helper T (T$_H$) cells and their subsets, including CXCR5$^+$CXCR3$^+$ type 1 T$_{FH}$ cells, CXCR5$^+$CXCR3$^-$ type 2/17 T$_{FH}$ cells, CXCR5$^-$CXCR3$^+$ type 1 T$_H$ cells and CXCR5$^-$CXCR3$^-$ type 2/17 T$_H$ cells increased at V3 and V4 compared to V1 in the FN and NI cohorts (Fig. 4i,j). However, we found reduced frequencies of CD134$^+$CD137$^+$ T$_{FH}$ cells, type 1 T$_{FH}$ cells and type 2/17 T$_{FH}$ cells in Australian FN peoples with CMs compared to FN peoples without CMs, but not in the T$_H$ subsets (Fig. 4k). NI individuals with IBD showed CD134$^+$CD137$^+$ T$_{FH}$ and T$_H$ cell responses comparable to NI individuals without CMs (Fig. 4k). The frequency of CD134$^+$CD137$^+$ T$_{FH}$ subsets and CD69$^+$CD137$^+$CD8$^+$ T cells at V3 correlated with RBD-specific IgG titers at V3 in FN participants but not in the T$_H$ subsets or CD4$^+$ T cells (Fig. 4l). Overall, reduced frequency of spike-specific T$_{FH}$ responses were observed in FN individuals with CMs than FN individuals without CMs.

## FN individuals have robust CD4$^+$ but low CD8$^+$Tet$^+$ T cell responses

We used peptide-HLA tetramers and tetramer-associated magnetic enrichment to define SARS-CoV-2 epitope-specific CD4$^+$ and CD8$^+$ T cell responses[29,30] at V1, V3, V4 and V5 in 17 FN and 13 NI Australians. Prominent and conserved epitopes included DPB1*04:01 (DPB4)/S$_{167}$ (refs. 15,30), A*01:01 (A1)/S$_{865}$, A*02:01 (A2)/S$_{269}$, A*03:01 (A3)/S$_{378}$ and A*24:02 (A24)/S$_{1,208}$ (refs. 29,31–34). HLA frequencies in Australian FN peoples or NI groups were well represented (Fig. 5a).

**Fig. 2 | Higher bulk IgG G0 and lower RBD IgG in FN individuals with CMs.** **a**, V3 ancestral RBD IgG correlations with age (FN: $n = 46$; NI: $n = 39$), BMI (FN: $n = 39$; NI, $n = 36$) and grouped by sex (FN: $n_{Male} = 22$, $n_{Female} = 24$; NI: $n_{Male} = 10$, $n_{Female} = 29$). **b,c**, Correlation between age and ancestral RBD IgG at V3 (**b**) and ELISA showing ancestral RBD IgG at V3 in individuals with or without CMs (WCMs) (**c**). FN: $n_{WCM} = 28$, $n_{CM} = 18$, WCM versus CM, $P = 0.0084$. **d**, Bulk IgG glycan profiling showing 0 (G0), 1 (G1) or 2 (G2) galactose(s) at V1 and G0 with core fucose (G0f) at V1 and V3 ($n_{FN} = 26$; V1 G2 FN versus NI, $P = 0.0001$). **e**, Glycan profiling showing G0 at V1, V3 and V5 (FN: $n_{V1} = 26$, $n_{V3} = 27$, $n_{V5} = 10$; NI: V1 versus V3, $P = 0.0144$). **f**, Correlation between G0 at V1 and ancestral RBD IgG at V3 ($n_{FN} = 26$). **g**, Glycan profiling showing G0, fucose and G2 with 1 sialylation and core fucose (G2S1f) at V1 in WCMs or CMs (FN: $n_{WCM} = 15$, $n_{CM} = 11$, fucose WCM versus CM, $P = 0.0150$). **h**, Correlation between age and G0 at V1 (FN: $n_{WCM} = 14$, $n_{CM} = 11$). **i**, ELISA showing the avidity of ancestral RBD IgG at V3 in FN, NI and WCM or CM (FN: $n_{WCM} = 28$, $n_{CM} = 18$). **j**, Correlation between ancestral RBD IgG and avidity at V3 ($n_{FN} = 46$). **k,l**, LEGENDplex assay showing concentrations of 13 cytokines and chemokines (**k**) and IL-18 (**l**) in WCM or CM FN and NI individuals at V1 (FN: $n_{WCM} = 23$, $n_{CM} = 14$; IL-18 WCM versus CM, $P = 0.0092$). The stacked bar graph shows the mean concentration. **m**, Correlation between IL-18 concentration at V1 with G0 abundance at V1 and RBD IgG at V3 in WCM or CM FN and NI individuals ($n_{FN} = 26$). **n**, Frequency of spike-specific B cells among IgD$^-$ B cells at V1, V3, V4 and V5 (FN: $n_{V1,V4} = 11$, $n_{V3} = 17$, $n_{V5} = 7$, V1 versus V3, $P = 0.0117$; NI: V1 versus V4, $P = 0.0007$; V4 FN versus NI, $P = 0.0419$). **o**, Frequency of spike-specific B cells among IgD$^-$ B cells in individuals with or without CMs at V4 (FN: $n_{WCM} = 7$, $n_{CM} = 4$, WCM versus CM, $P = 0.0242$). **p**, Correlations between the frequency of spike-specific B cells among IgD$^-$ B cells and ancestral RBD IgG at V3 ($n_{FN} = 17$) or V4 ($n_{FN} = 11$). In **b,c,i,j** $n_{NI} = 39$. In **d–h,k–m** $n_{NI} = 33$. In **n–p** $n_{NI} = 13$. The bars indicate the median with the IQR. The horizontal dotted line indicates seropositivity. Statistical significance was determined with a two-sided Wilcoxon test within cohort or two-sided Mann–Whitney $U$-test between cohorts. *$P < 0.05$, **$P < 0.01$, ***$P < 0.001$, ****$P < 0.0001$. Correlation was determined using Spearman correlation.

Two mRNA vaccine doses elicited robust CD4[+15] and CD8[+] T cell responses[14,16] in NI peoples. Ex vivo CD4[+] T cells directed at DPB4/$S_{167}$ were of a similar magnitude in FN and NI Australians at V3 (Fig. 5b–d and Extended Data Fig. 2c), confirming the immunodominance of DPB4/$S_{167}$[+]CD4[+] T cells. Pooled SARS-CoV-2-specific CD8[+] T cell responses increased at V3 than V1 in FN peoples but were lower than those detected in NI participants (Fig. 5b–d). Similar naive precursor frequencies of DPB4/$S_{167}$[+]CD4[+] T cells were detected at V1 between the FN and NI cohorts (Fig. 5e). Precursor frequency of

epitope-specific CD8[+] T cells can affect the magnitude of CD8[+] T cell responses[29,35]. The frequencies of Tet[+]CD8[+] T cells specific for the A1/$S_{865}$, A3/$S_{378}$ or A24/$S_{1,208}$ epitopes were similar at V1 in the FN and NI cohorts, while the frequency of A2/$S_{269}$[+]CD8[+] T cells was higher in Australian FN peoples than in NI peoples at V1 (Fig. 5e), suggesting that lower SARS-CoV-2-specific CD8[+] T cell responses in Australian FN participants after vaccination were not due to lower precursor frequencies at V1. An increase in DPB4/$S_{167}$[+]CD4[+] T cells was observed at V3 compared to V1 in the FN and NI cohorts (Fig. 5f). The frequency

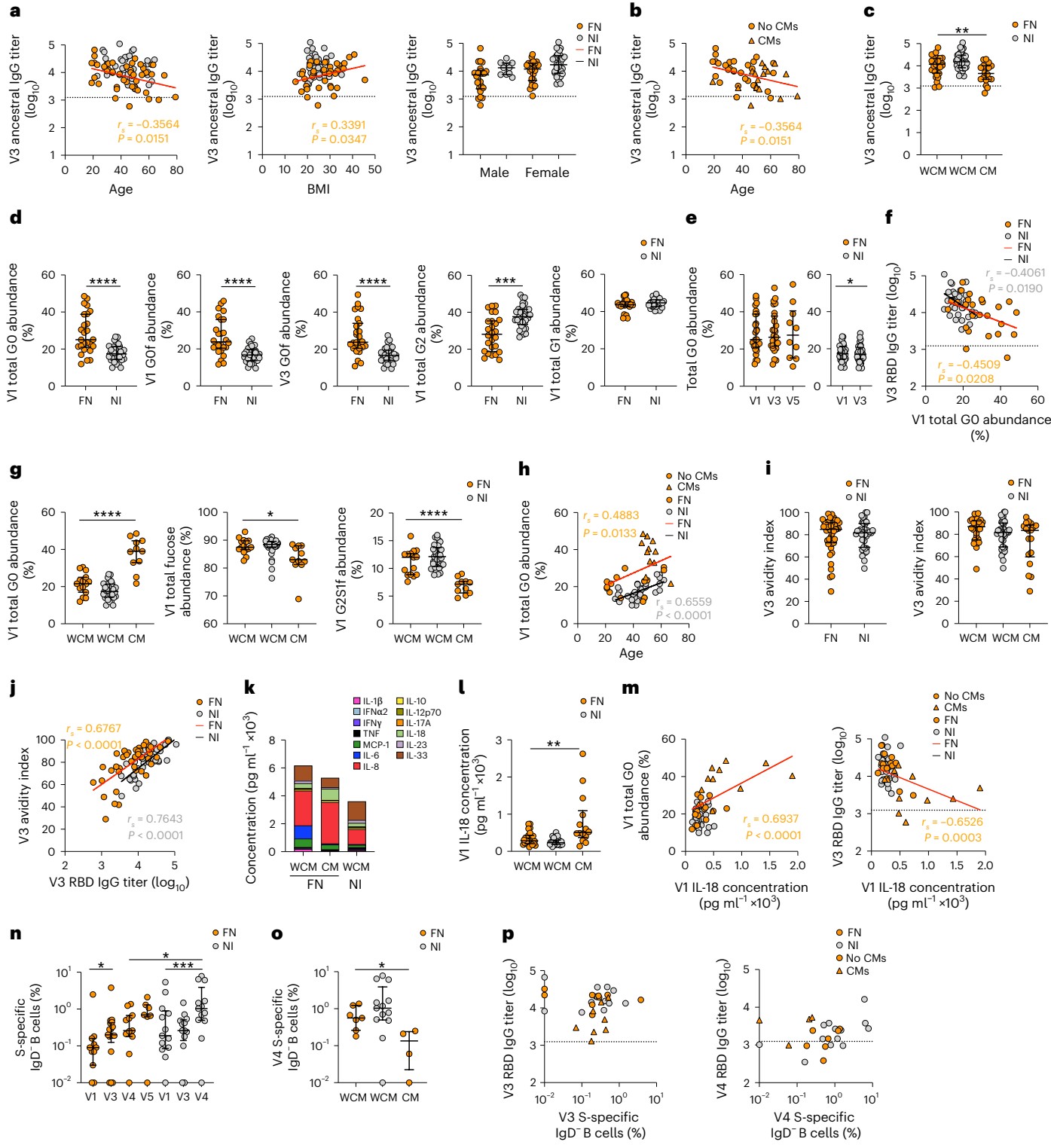

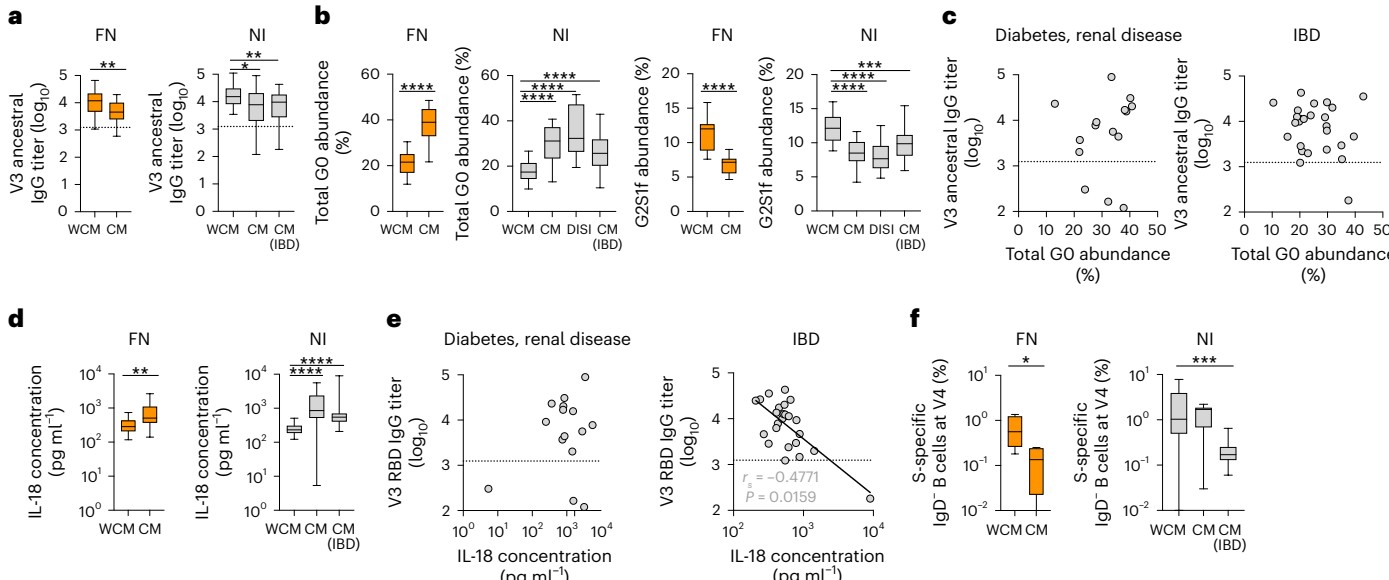

**Fig. 3 | Reduced RBD-specific IgG titers and spike-specific B cells with increased global IgG G0 abundance in NI people with CMs.** **a**, ELISA showing ancestral RBD IgG at V3 for FN and NI individuals with or without CMs and NI individuals with IBD. FN WCM versus CM, $P = 0.0084$; NI: WCM versus CM, $P = 0.0360$; NI: WCM versus IBD, $P = 0.0058$. **b**, Glycan profiling showing abundance of total 0 galactose (G0) and 2 galactoses with 1 sialylation and core fucose (G2S1f). **c**, Spearman correlation between total G0 and ancestral RBD IgG at V3. **d**, LEGENDplex assay showing IL-18 concentrations in FN and NI individuals with or without CMs. FN: WCM versus CM, $P = 0.0092$. **e**, Spearman correlation between IL-18 concentrations and ancestral RBD IgG at V3. FN: $n_{WCM} = 28$, $n_{CM} = 18$; NI: $n_{WCM} = 39$, $n_{CM} = 15$, $n_{DISI} = 16$, $n_{IBD} = 26$. The black horizontal dotted line indicates seropositivity. **f**, Frequency of spike-specific B cells among IgD$^-$ B cells at V4. FN: $n_{WCM} = 7$, $n_{CM} = 4$; NI: $n_{WCM} = 13$, $n_{CM} = 8$, $n_{IBD} = 23$. FN: WCM versus CM, $P = 0.0242$; NI: WCM versus IBD, $P = 0.0002$. The whiskers of the box plots indicate the minima and maxima, the bounds of the box indicate the IQR and the bars indicate the median. Statistical significance was determined with a two-sided Mann–Whitney $U$-test. *$P < 0.05$, **$P < 0.01$, ***$P < 0.001$, ****$P < 0.0001$. Samples from NI individuals with CMs are detailed in the Methods.

of A2/S$_{269}$$^+$CD8$^+$ T cells and, to a lesser extent, that of A24/S$_{1,208}$$^+$CD8$^+$ T cells increased in NI individuals (Fig. 5f), suggesting reduced expansion of these epitope-specific CD8$^+$ T cells in Australian FN peoples at V3. This could be due to the A1/S$_{865}$, A2/S$_{269}$, A3/S$_{378}$ or A24/S$_{1,208}$ epitopes being identified in NI peoples that may not be as immunodominant in Australian FN peoples expressing different HLA class I glycoproteins[36] (Supplementary Table 3).

SARS-CoV-2-specific CD27$^+$CD45RA$^-$ activated central memory (T$_{CM}$)-like CD8$^+$ T cells increased while CD27$^+$CD45RA$^+$CCR7$^+$CD95$^-$ naive CD8$^+$ T cells decreased at V3, V4 or V5 compared to V1 in both Australian FN and NI peoples (Fig. 5g,h). The frequencies of Tet-specific CD27$^+$CD45RA$^-$ T$_{CM}$-like CD4$^+$ T cells were similar at V3, V4 and V5 in Australian FN and NI participants (Fig. 5g,h). DPB4/S$_{167}$$^+$CD4$^+$ T cell numbers at V1 were insufficient for phenotyping analyses. Therefore, comparable DPB4/S$_{167}$$^+$CD4$^+$ but reduced Tet$^+$CD8$^+$ T cell frequencies were observed in FN peoples.

**Tet$^+$ T cells display prominent gene segment usage in FN**

T cell receptor (TCR) αβ clonal diversity and composition can affect T cell functionality and protection[37,38]. We dissected ex vivo SARS-CoV-2-specific TCRαβ repertoires for CD8$^+$ (A2/S$_{269}$ and A24/S$_{1,208}$) and CD4$^+$ (DPB4/S$_{167}$) T cell epitopes from 11 Australian vaccinated FN individuals (no reported SARS-CoV-2 infection) and 6 infected with SARS-CoV-2 (received 2–3 mRNA vaccine doses) all hospitalized (median age of 61 (range 33–69); Supplementary Table 4). Samples were collected at hospital admission (median 4 days after disease onset) and before discharge (median 11 days after disease onset). In total, 529 SARS-CoV-2-specific TCRs (334 paired TCR clonotypes; Supplementary Table 5) from Australian FN peoples were compared to published TCR datasets from NI adults and children with SARS-CoV-2 infection[29,30,33]. The TCRαβ repertoires for DPB4/S$_{167}$$^+$CD4$^+$ T cells (Fig. 6a), A24/S$_{1,208}$$^+$CD8$^+$

T cells (Fig. 6b) and A2/S$_{269}$$^+$CD8$^+$ T cells (Fig. 6c) displayed similarity between vaccinated and SARS-CoV-2-infected Australian FN peoples and SARS-CoV-2-infected NI individuals (Fig. 6). Comparable to NI peoples[15,30], DPB4/S$_{167}$$^+$CD4$^+$ T cells displayed a heavy bias for *TRAV35*/*TRAJ42* gene segments, along with a prominent 'CXXXNYGGSQGNLIF' complementarity-determining region (CDR) 3α motif (X denotes any amino acid), and accounted for most of the TCRα repertoire in FN peoples after COVID-19 vaccination or SARS-CoV-2 infection (Fig. 6a). DPB4/S$_{167}$ CDR3β 'CASSLRGDTQYF' produced by *TRBV11-2*/*TRBJ2-3* was shared between FN and NI peoples (Fig. 6a).

The A24/S$_{1,208}$$^+$CD8$^+$ T cell TCRαβ repertoire was extremely diverse and private (not shared between individuals) in NI[33] and FN peoples (Fig. 6b). Despite this, we found common *TRAV* and *TRBV* gene segments (*TRAV16*, *TRBV5-6*, *TRBV20-1*) detected at similar frequencies in Australian FN peoples and NI peoples (Fig. 6b). The A2/S$_{269}$$^+$CD8$^+$ TCR repertoires in FN peoples were also highly diverse (Fig. 6c), in contrast to ex vivo TCR data in NI adults and children, who generally have a more public (although relatively diverse) repertoire, with biased *TRAV12-1*/*TRAJ43* and *TRAV12-2*/*TRAJ30* gene usage and conserved CDR3 motifs (CVVNXXXDMRF and CAVNXDDKIIF, respectively)[29,30,39]. These *TRAV* genes and motifs were minimally observed in FN peoples (Fig. 6c).

Using a published NI adult database[29,30], we tested whether certain V segments were enriched or depleted in FN and NI peoples within A2/S$_{269}$$^+$CD8$^+$ TCRs. A Fisher exact test indicated that the *TRAV12-1*01* frequency within the FN response was statistically distinct from NI adults (Extended Data Fig. 4a). We observed an enriched TCR α-chain motif in the NI A2/S$_{269}$ response, largely driven by *TRAV12-1* usage, which was not observed in the FN A2/S$_{269}$ TCR repertoire (Extended Data Fig. 4b,c). This difference resulted in a TCR α-chain A2/S$_{269}$ repertoire that was very diverse in FN compared to NI peoples (Extended Data

Fig. 4d). Similar A2/$S_{269}$⁺CD8⁺ TCR β-chain motifs were observed in FN and NI participants, largely driven by *TRBJ2-2* (Extended Data Fig. 4b–d), suggesting that the core features of the A2/$S_{269}$-specific TCR repertoire are preserved in both populations. Analysis of public bulk RNA sequencing datasets from three FN Australians[40] showed no mutations in germline CDR1 or CDR2 regions or other parts of the *TRAV12-1* sequences; usage of this germline segment in bulk TCR repertoire was similar to datasets[41]. Although common TCRαβ repertoire features exist between Australian FN and NI peoples within immunodominant DPB4/$S_{167}$⁺CD4⁺ and A24/$S_{1,208}$⁺CD8⁺ T cells, the A2/$S_{269}$⁺CD8⁺ T cell repertoire of Australian FN peoples was highly diverse and lacked publicly reported gene motifs.

## Decreased SARS-CoV-2 antibodies are linked to altered IgG glycosylation

We generated correlation matrices for Australian FN individuals with and without CMs, NI individuals without CMs and FN individuals with CMs between clinical features and immune parameters at V3. Age was negatively correlated with the abundance of total G2 glycosylation, and abundance of G2S1f glycosylation in the FN and NI cohorts (Fig. 7a,b), confirming the reports in NI cohorts[42]. For FN peoples, age was negatively correlated with RBD IgG avidity and microneutralization titer (Fig. 7a). The ancestral RBD IgG titer correlated with IFNα2 and IFNγ, but was negatively correlated with IL-18 in Australian FN individuals with and without CMs (Fig. 7a). The frequencies of CD69⁺CD137⁺CD8⁺ T cells correlated with IFNγ and IL-33, whereas the frequencies of IFNγ⁺TNF⁺CD4⁺ T cells correlated with IL-1β, MCP-1 and IL-6 in FN individuals with and without CMs (Fig. 7a), but not in NI peoples or Australian FN peoples with CMs (Fig. 7b,c). There were differences in bulk IgG glycosylation and cytokine measurements between Australian FN individuals with and without CMs (Fig. 7d). FN individuals with CMs had higher bulk IgG G0f and total G0 compared to FN individuals without CMs, while ancestral RBD IgG and microneutralization titers were higher in FN individuals without CMs (Fig. 7d). Overall, we found a reduced SARS-CoV-2 antibody axis linked to altered IgG glycosylation levels after COVID-19 vaccination in Australian FN peoples with CMs.

## Discussion

We provide key insights into immune responses after COVID-19 vaccination in Indigenous people, link antibody glycosylation levels to reduced antibody titers after COVID-19 vaccination in any population and emphasize the importance of vaccine-induced T cells in individuals with CMs.

Generation of SARS-CoV-2-specific CD8⁺ T cells by mRNA vaccination is of importance because many vaccines, including seasonal influenza inactivated vaccines[43], do not elicit virus-specific CD8⁺ T cells. Establishment of memory T cells, especially CD8⁺ T cells, is important

because T cells are mainly directed toward epitopes encompassing conserved viral peptides; thus, they can respond to SARS-CoV-2 variants, including Omicron[17]. T cells also have a role in limiting COVID-19 severity in immunosuppressed individuals lacking B cells and antibodies[44].

Our data showed prominent and comparable SARS-CoV-2-specific T cell responses by AIM and ICS in Australian FN and NI peoples after mRNA vaccination, irrespectively of CMs. People with CMs have robust T cell responses after COVID-19 vaccination, suggesting some level of protection for subsequent SARS-CoV-2 infection. This included CD4⁺ T cell responses directed against the prominent DPB4/$S_{167}$ epitope[15], suggesting immunodominance of DPB4/$S_{167}$⁺CD4⁺ T cells in FN and NI peoples.

In contrast to AIM and ICS T cell responses, tetramer-specific CD8⁺ T cell responses (A1/$S_{865}$, A2/$S_{269}$, A3/$S_{378}$, A24/$S_{1,208}$) previously identified as immunodominant in NI peoples[29–31,33], were reduced in FN peoples. Differential HLA expression profiles in FN peoples[36] could explain why these epitopes were not immunodominant in FN individuals. Understanding the immunogenicity and magnitude hierarchy of SARS-CoV-2-epitopes restricted by HLAs prominent in FN peoples is of importance for future studies, as exemplified by our work for influenza viruses[45,46].

Although RBD antibody levels increased reaching approximately 92% seroconversion in FN Australians after two mRNA vaccine doses, antibody titers in FN peoples were lower than NI peoples, which was attributed to chronic CMs in FN peoples. The whole antibody-producing axis, including B cells and type 1 T_FH cells, was reduced in Australian FN peoples with CMs after two vaccine doses, making those individuals more susceptible to severe COVID-19, especially with newly emerging VOCs. Altered IgG glycosylation levels in all participants with CMs were associated with lower RBD-specific IgG titers and increased before-vaccination IL-18 (increased in severe COVID-19 (ref. 25)). Our observations suggests that IgG G0 abundance (potentially together with IL-18) before vaccination might serve as a biomarker for humoral responses after vaccination for COVID-19 or other infectious diseases.

While our data demonstrate altered IgG G0 glycosylation levels in all participants, these findings are highly relevant to Australian FN peoples because the burden of chronic conditions is considerably higher in this population[47]. Age-standardized years of life lost from chronic conditions in the Northern Territory FN population is 4.06 times higher than Australia as a whole[47], representing a 15-year reduction in life expectancy for Australian FN men and a 19-year reduction for FN women when comparing NI men and women living in the Northern Territory[48]. Chronic renal disease and diabetes mellitus have a disproportionate prevalence in the younger Northern Territory FN population than NI peoples[49,50], mirroring other Indigenous populations globally.

TCRαβ analysis showed that DPB4/$S_{167}$⁺CD4⁺ T cells from vaccinated FN and NI peoples shared a common *TRAV35/TRAJ43* CDR3α

---

**Fig. 4 | Prototypical CD4⁺ and CD8⁺ T cell responses in FN and NI cohorts.**
**a–c**, Representative flow cytometry plots (**a**) and frequency (**b,c**) of CD134⁺CD137⁺CD4⁺ T cells (FN: V1 versus V4, $P = 0.0020$) and CD69⁺CD137⁺CD8⁺ T cells (FN: V1 versus V3, $P = 0.0007$; NI: V1 versus V4, $P = 0.0027$; V3 FN versus NI, $P = 0.0496$) at V1, V3 and V4 (FN: $n_{V1} = 31$, $n_{V3} = 41$, $n_{V4} = 21$; NI: $n_{V1} = 32$, $n_{V3} = 34$, $n_{V4} = 25$) (**b**) and V1, V3, V4 and V5 ($n_{V5} = 10$) (**c**). **d–f**, Representative flow cytometry plots (**d**) and frequency (**e,f**) of IFNγ⁺TNF⁺CD4⁺ T cells (FN: V1 versus V3, $P = 0.0003$; FN: V1 versus V4, $P = 0.0098$; V3 FN versus NI, $P = 0.0496$) and IFNγ⁺TNF⁺CD8⁺ T cells (FN: V1 versus V3, $P = 0.0335$) at V1, V3 and V4 (FN: $n_{V1} = 29$, $n_{V3} = 41$, $n_{V4} = 21$; NI: $n_{V1} = 32$, $n_{V3} = 34$, $n_{V4} = 25$) (**e**) and V1, V3, V4 and V5 ($n_{V5} = 9$) (**f**). **g,h**, Frequency of CD134⁺CD137⁺CD4⁺ T cells and CD69⁺CD137⁺CD8⁺ T cells (NI: WCM versus IBD, $P = 0.0384$) (**g**), and IFNγ⁺TNF⁺CD4⁺ T cells and IFNγ⁺TNF⁺CD8⁺ T cells (**h**) in individuals with CMs or WCMs (FN: $n_{WCM} = 26$, $n_{CM} = 15$; NI: $n_{WCM} = 34$, $n_{CM} = 13$, $n_{IBD} = 26$). **i–k**, Representative flow cytometry plots (**i**) and frequency (**j,k**) of CD134⁺CD137⁺CXCR5⁺CD4⁺ T_FH cells (FN: V1 versus V4, $P = 0.0020$), CXCR5⁺CXCR3⁺ type 1 T_FH cells (FN: V1 versus V4, $P = 0.0195$; NI: V1 versus V4,

$P = 0.0001$; V1 FN versus NI, $P = 0.0382$), CXCR5⁺CXCR3⁻ type 2/17 T_FH cells (FN: V1 versus V3, $P = 0.0001$; FN: V1 versus V4, $P = 0.0029$; V3 FN versus NI, $P = 0.0480$), CXCR5⁻CD4⁺ T_H cells; FN: V1 versus V4, $P = 0.0020$), CXCR5⁻CXCR3⁺ type 1 T_H cells (FN: V1 versus V4, $P = 0.0020$; NI: V1 versus V4, $P = 0.0004$) and CXCR5⁻CXCR3⁻ type 2/17 T_H cells (FN: V1 versus V4, $P = 0.0020$; NI: V1 versus V4, $P = 0.0004$) at V1, V3 and V4 (FN: $n_{V1} = 31$, $n_{V3} = 41$, $n_{V4} = 21$; NI: $n_{V1} = 32$, $n_{V3} = 34$, $n_{V4} = 25$) (**j**) in individuals with CMs or WCMs at V3 (FN: $n_{WCM} = 26$, $n_{CM} = 15$; NI: $n_{WCM} = 34$, $n_{CM} = 13$, $n_{IBD} = 26$; T_FH cells: FN: WCM versus CM, $P = 0.0276$; type 1 T_FH cells: FN: WCM versus CM, $P = 0.0156$; type 2/17 T_FH cells: FN CM versus NI WCM, $P = 0.0150$) (**k**). **l**, Spearman correlations between ancestral RBD IgG and CD134⁺CD137⁺ T_FH, type 1 T_FH, type 2/17 T_FH, T_H, type 1 T_H, type 2/17 T_H, CD4⁺ and CD69⁺CD137⁺CD8⁺ T cells at V3 ($n_{FN,NI} = 33$). The bars indicate the median with the IQR. Statistical significance was determined with a two-sided Wilcoxon test in each cohort or two-sided Mann–Whitney *U*-test between FN and NI cohorts. *$P < 0.05$, **$P < 0.01$, ***$P < 0.001$, ****$P < 0.0001$.

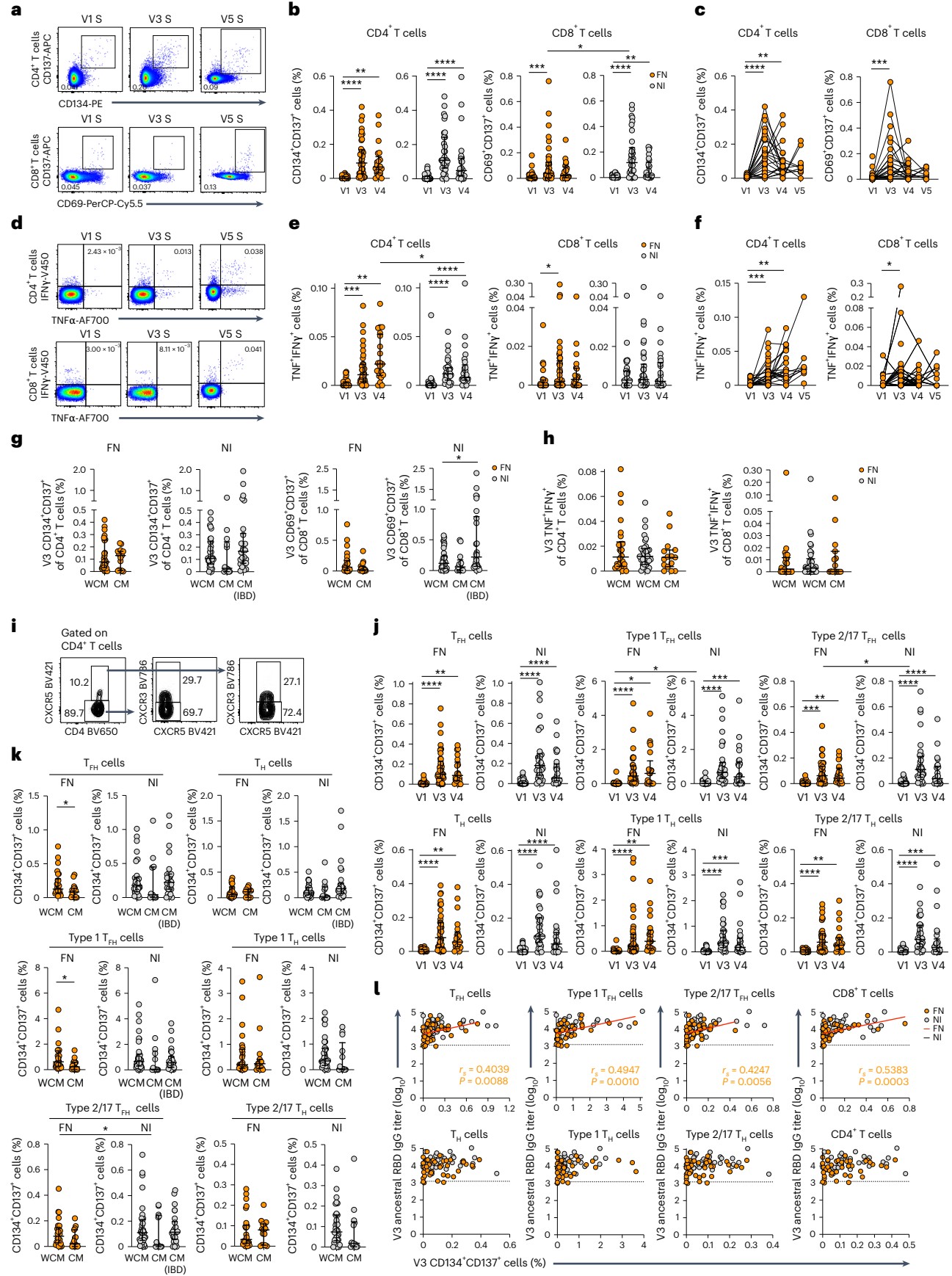

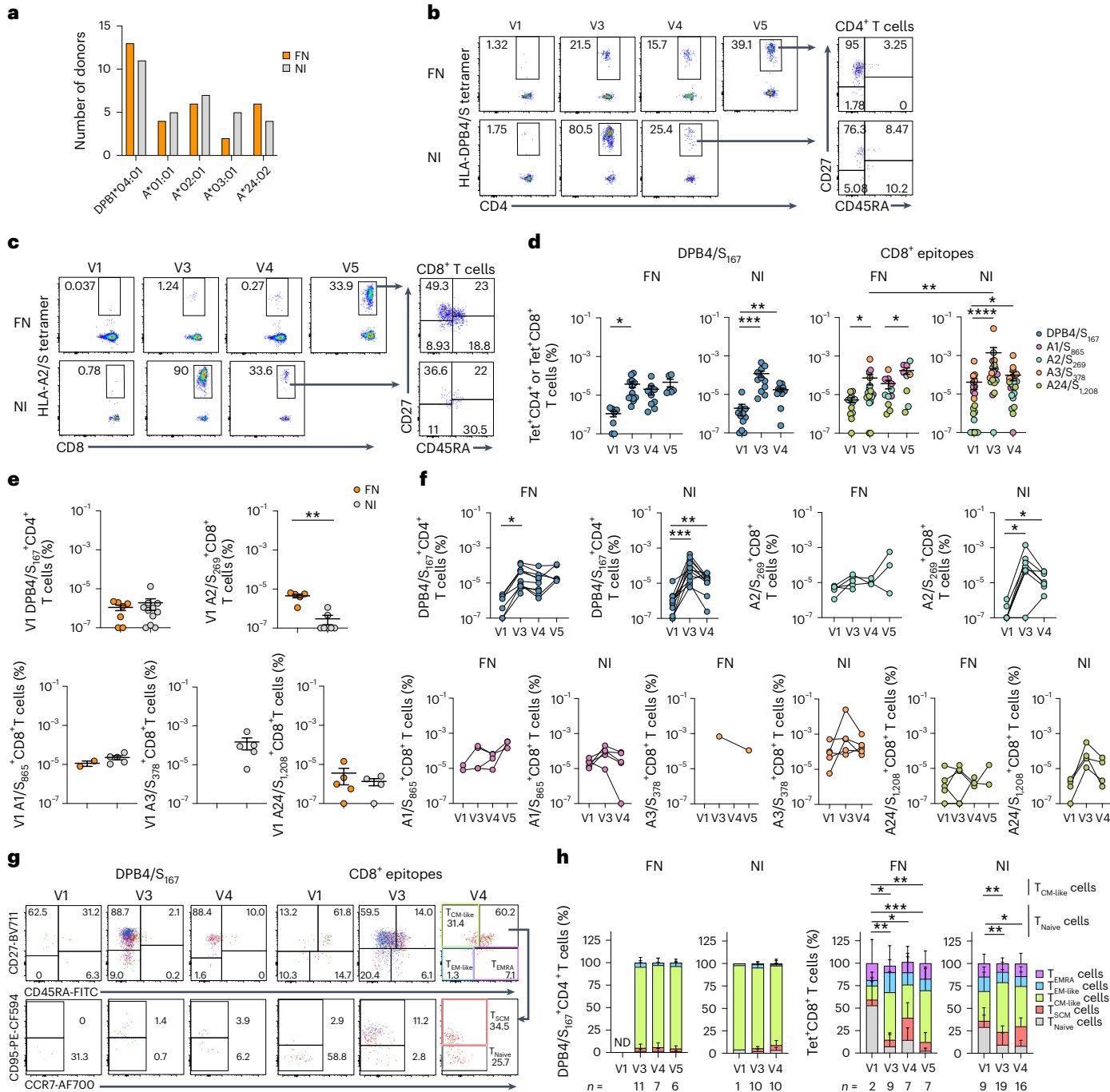

**Fig. 5 | Higher tetramer⁺CD8⁺ T cells in the NI cohort than in the FN cohort.**
**a**, Bar graph showing the number of donors with each HLA of interest in the FN and NI samples. **b**–**d**, Representative flow cytometry plots (**b**,**c**) and frequency (**d**) of Tet(DPB4/$S_{167}$)⁺CD4⁺ (FN: $n_{V1}$ = 7, $n_{V3}$ = 13, $n_{V4}$ = 9, $n_{V5}$ = 6, V1 versus V3, $P$ = 0.0156; NI: $n_{V1,V3,V4}$ = 11, V1 versus V3, $P$ = 0.0010, V1 versus V4, $P$ = 0.0020) (**b**,**d**) and Tet(A1/$S_{865}$ or A2/$S_{269}$ or A3/$S_{378}$ or A24/$S_{1,208}$)⁺CD8⁺ T cells (FN: $n_{V1}$ = 12, $n_{V3}$ = 17, $n_{V4}$ = 11, $n_{V5}$ = 9, V1 versus V3, $P$ = 0.0322, V4 versus V5, $P$ = 0.0391; NI: $n_{V1,V3,V4}$ = 21, V1 versus V4, $P$ = 0.0101; V3 FN versus NI, $P$ = 0.0026) (**c**,**d**) at V1, V3, V4 and V5 in the FN and NI samples. **e**, Frequency of Tet⁺CD4⁺ and Tet⁺CD8⁺ T cells (DPB4/$S_{167}$: $n_{FN}$ = 7, $n_{NI}$ = 11; A2/$S_{269}$: $n_{FN}$ = 5, $n_{NI}$ = 7, $P$ = 0.0038; A1/$S_{865}$: $n_{FN}$ = 2, $n_{NI}$ = 5; A3/$S_{378}$: $n_{NI}$ = 5; A24/$S_{1,208}$: $n_{FN}$ = 5, $n_{NI}$ = 4) at V1 in the FN and NI samples. **f**, Frequency of Tet⁺CD4⁺ and Tet⁺CD8⁺ T cells for each peptide-HLA tetramer at V1, V3, V4 and V5 (DPB4/$S_{167}$: FN: $n_{V1}$ = 7, $n_{V3}$ = 11, $n_{V4}$ = 9, $n_{V5}$ = 6, V1 versus V3, $P$ = 0.0156; NI: $n_{V1}$ = 1, $n_{V3}$ = 11, $n_{V4}$ = 11, V1 versus V3, $P$ = 0.0010, NI: V1 versus V4, $P$ = 0.0020; A2/$S_{269}$: FN: $n_{V1}$ = 5, $n_{V3}$ = 5, $n_{V4}$ = 3, $n_{V5}$ = 3; NI: $n_{V1}$ = 7, $n_{V3}$ = 7, $n_{V4}$ = 7, V1 versus V3, $P$ = 0.0312, V1 versus V4, $P$ = 0.0156; A1/$S_{865}$: FN: $n_{V1}$ = 2, $n_{V3}$ = 4, $n_{V4}$ = 4, $n_{V5}$ = 3; NI: $n_{V1}$ = 5, $n_{V3}$ = 5, $n_{V4}$ = 5; A3/$S_{378}$: FN: $n_{V3}$ = 1, $n_{V5}$ = 1; NI: $n_{V1}$ = 5, $n_{V3}$ = 5, $n_{V4}$ = 5;

A24/$S_{1,208}$: FN: $n_{V1}$ = 5, $n_{V3}$ = 5, $n_{V4}$ = 3, $n_{V5}$ = 2; NI: $n_{V1,V3,V4}$ = 4). The bars indicate the mean ± s.e.m. Statistical significance was determined with a two-sided Wilcoxon test in each cohort or a Mann–Whitney $U$-test between FN and NI cohorts. **g**,**h**, Representative flow cytometry (**g**) and stacked bar graphs (**h**) showing concatenated representative plots of the Tet⁺CD4⁺ T cell and Tet⁺CD8⁺ T cell phenotypes (CD27⁺CD45RA⁺CD95⁻CCR7⁺ $T_{Naïve}$ cells, CD27⁺CD45RA⁺CD95⁺ stem cell memory T ($T_{SCM}$) cells, CD27⁺CD45RA⁻ $T_{CM-like}$ cells, CD27⁻CD45RA⁻ $T_{EM-like}$ cells, CD27⁻CD45RA⁺ terminally differentiated effector memory ($T_{EMRA}$) cells). Only samples of 10 or more tetramer⁺-enriched events were included for analysis (DPB4/$S_{167}$: FN: $n_{V3}$ = 11, $n_{V4}$ = 7, $n_{V5}$ = 6; NI: $n_{V1}$ = 1, $n_{V3}$ = 10, $n_{V4}$ = 10; CD8⁺ epitopes: FN: $n_{V1}$ = 2, $n_{V3}$ = 9, $n_{V4}$ = 7, $n_{V5}$ = 7; $T_{Naïve}$: V1 versus V3, $P$ = 0.0013; V1 versus V4, $P$ = 0.0173; V1 versus V5, $P$ = 0.0004; $T_{CM-like}$: V1 versus V3, $P$ = 0.0173; V1 versus V5, $P$ = 0.0065; NI: $n_{V1}$ = 9, $n_{V3}$ = 19, $n_{V4}$ = 16; $T_{Naïve}$: V1 versus V3, $P$ = 0.0100; V1 versus V4, $P$ = 0.0150; $T_{CM-like}$: V1 versus V3, $P$ = 0.0035). ND, not determined. The mean and s.d. are shown in the stacked plots. Statistical significance was determined with a two-way ANOVA with Tukey's multiple comparisons. *$P$ < 0.05, **$P$ < 0.01, ***$P$ < 0.001, ****$P$ < 0.0001.

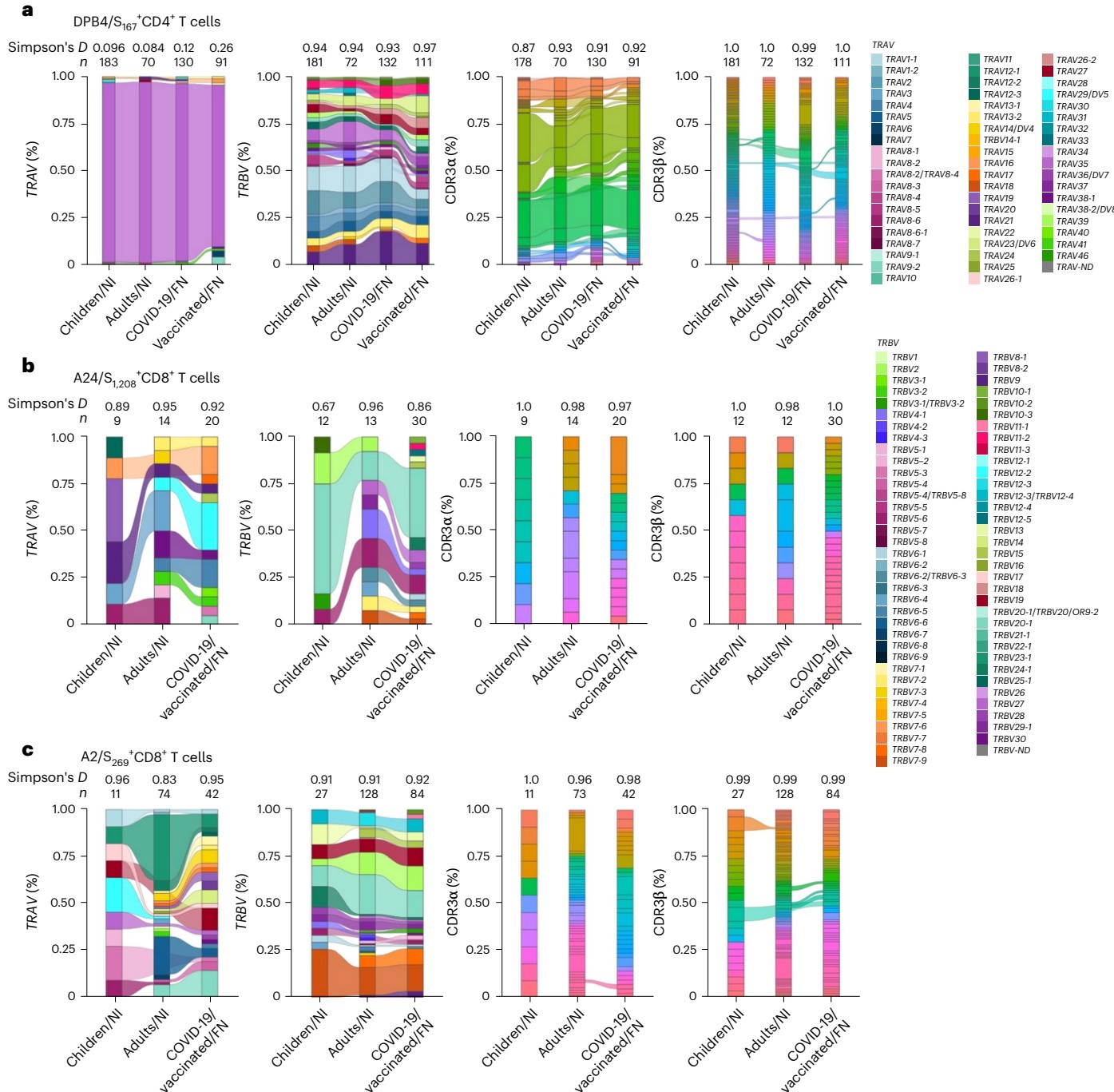

**Fig. 6 | *TRAV/TRBV* and CDR3α/β usage in FN and NI individuals. a–c,** Alluvial plots showing the frequency of *TRAV/TRBV* gene usage and CDR3α/β clonotypes in DPB4/S$_{167}$-specific (**a**), A24/S$_{1,208}$-specific (**b**) and A2/S$_{269}$-specific (**c**) TCRs from NI children and adults convalescing from COVID-19 (refs. 29,30,33), FN adult individuals with COVID-19 and vaccinated FN adults (for the A2 and A24 plots, FN individuals with COVID-19 and vaccinated adults were pooled). The connections between the bars represent CDR3 clonotypes shared between individuals in different cohorts. Full TCR sequences for FN adults are listed in Supplementary Table 5. Full CDR3α and β legends are listed in Extended Data Fig. 3.

motif (CXXXNYGGSQGNLIF). TCR repertoires for A24/S$_{1,208}$$^+$CD8$^+$ and A2/S$_{269}$$^+$CD8$^+$ T cells were highly diverse in Australian FN peoples, similar to NI peoples[30,33]. However, within the A2/S$_{269}$$^+$CD8$^+$ TCR repertoire of Australian FN peoples, lack of two public motifs may explain partially why there was little expansion of A2/S$_{269}$$^+$CD8$^+$ T cells after two mRNA vaccine doses, despite normal precursor frequency. Our data suggest a unique and diverse set of A2/S$_{269}$$^+$CD8$^+$ T cells, perhaps representing a unique TCR repertoire for Australian FN peoples, which may have functional consequences and requires further exploration.

This study is limited by epitope-specific T cell data focused on common HLAs, the cohorts originating from one country and the heterogenous NI cohort with CMs with respect to the timing of sample collection after COVID-19 vaccination. Our study provides an immunological basis to support current vaccine recommendations in Indigenous populations and has important implications for vaccination regimens in individuals with chronic conditions, including diabetes and renal disease. COVID-19 vaccination schedules, including boosters, monoclonal antibodies and immunomodulatory therapies,

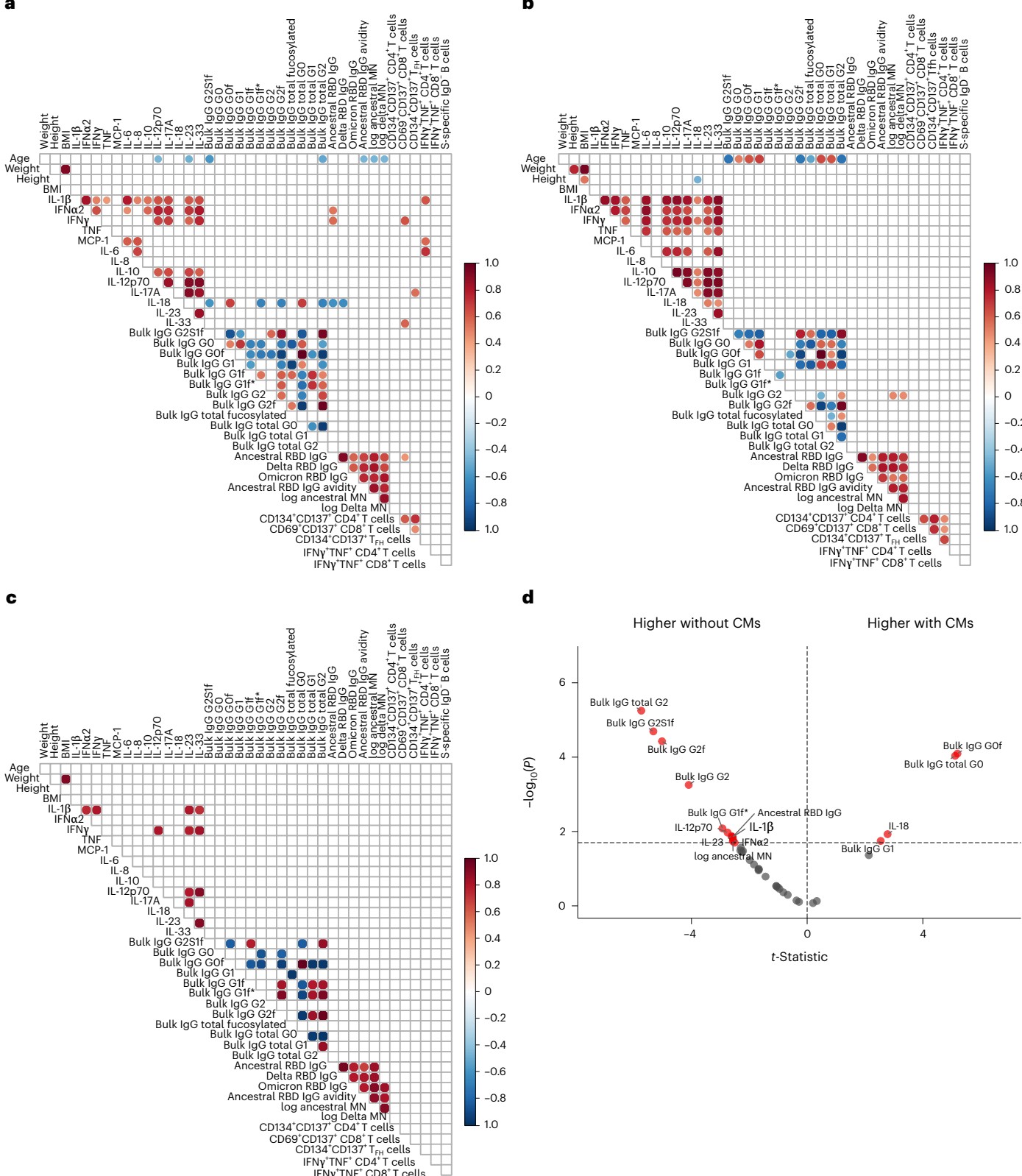

**Fig. 7 | Correlations between clinical and immune features. a–c,** Correlation matrices showing the Spearman correlations of clinical features, serological and cellular immune responses for FN individuals with or without CMs (**a**), NI individuals without CMs (**b**) and FN individuals with CMs (**c**). **d,** Volcano plot comparing 37 clinical and immunological features between FN individuals with and without CMs. MN, microneutralization. G1f and G1f* indicate positional isomers. Statistical significance was determined with an unpaired, two-sided *t*-test with Benjamini–Hochberg adjustment.

may be important considerations for individuals with substantial chronic conditions. Our study supports Australian recommendations for earlier or additional (or both) booster doses of COVID vaccines for high-risk groups, including those with chronic CMs and Indigenous people.

## Online content

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

[1]Department of Microbiology and Immunology, University of Melbourne, at the Peter Doherty Institute for Infection and Immunity, Parkville, Victoria, Australia. [2]Faculty of Veterinary and Agricultural Sciences, University of Melbourne, Melbourne, Victoria, Australia. [3]Menzies School of Health Research, Darwin, Northern Territory, Australia. [4]Infection and Immunity Program and Department of Biochemistry and Molecular Biology, Biomedicine Discovery Institute, Monash University, Clayton, Victoria, Australia. [5]Department of Immunology, St. Jude Children's Research Hospital, Memphis, TN, USA. [6]Central and Northern Adelaide Renal and Transplantation Service, Royal Adelaide Hospital, Adelaide, South Australia, Australia. [7]Adelaide Medical School, University of Adelaide, Adelaide, South Australia, Australia. [8]Department of Gastroenterology, Royal Melbourne Hospital, Melbourne, Victoria, Australia. [9]Macquarie University, Sydney, New South Wales, Australia. [10]Department of Biochemistry and Chemistry, La Trobe Institute for Molecular Science, La Trobe University, Bundoora, Victoria, Australia. [11]Department of Infectious Diseases, Peter Doherty Institute for Infection and Immunity, University of Melbourne and Royal Melbourne Hospital, Melbourne, Victoria, Australia. [12]Kirby Institute, University of New South Wales, Sydney, New South Wales, Australia. [13]Department of Infectious Diseases, Austin Health, Heidelberg, Victoria, Australia. [14]Victorian Infectious Diseases Services, Royal Melbourne Hospital and Doherty Department, University of Melbourne, Peter Doherty Institute for Infection and Immunity, Melbourne, Victoria, Australia. [15]World Health Organization Collaborating Centre for Reference and Research on Influenza, Peter Doherty Institute for Infection and Immunity, Melbourne, Victoria, Australia. [16]Department of Microbiology, Icahn School of Medicine at Mount Sinai, New York, NY, USA. [17]Department of Infectious Diseases, Alfred Hospital and Central Clinical School and School of Public Health and Preventive Medicine, Monash University, Melbourne, Victoria, Australia. [18]Monash Infectious Diseases, Monash Health and School of Clinical Sciences, Monash University, Melbourne, Victoria, Australia. [19]Department of Medicine, University of Melbourne, Parkville, Victoria, Australia. [20]Australian Research Council Centre of Excellence in Convergent Bio-Nano Science and Technology, University of Melbourne, Melbourne, Victoria, Australia. [21]Melbourne Sexual Health Centre, Infectious Diseases Department, Alfred Health, Central Clinical School, Monash University, Melbourne, Victoria, Australia. [22]Institute of Infection and Immunity, School of Medicine, Cardiff University, Cardiff, UK. [23]Central Australian Aboriginal Congress, Alice Springs, Northern Territory, Australia. [24]Indigenous Engagement, CQUniversity, Townsville, Queensland, Australia. [25]Infectious Diseases Department, Royal Darwin Hospital and Northern Territory Medical Programme, Darwin, Northern Territory, Australia. [26]Center for Influenza Disease and Emergence Response, Melbourne, Victoria, Australia. [27]These authors contributed equally: Thi H. O. Nguyen, Jane Davies, Katherine Kedzierska. ✉e-mail: tho.nguyen@unimelb.edu.au; jane.davies@menzies.edu.au; kkedz@unimelb.edu.au

## Methods

### COVAC study participants and specimens

We enrolled 97 participants that received the mRNA vaccine (58 Australian FN and 39 NI individuals; Fig. 1a) via the Menzies School of Health Research in Darwin, Northern Territory, Australia. Samples of vaccinees were taken at visit (V) 1 (before dose 1, $n = 71$), V1a (day 6–28 after dose 1, $n = 49$), V2 (before dose 2, $n = 75$), V3 (day 28 after dose 2, $n = 85$), V4 (6 months after dose 2, $n = 57$) and V5 (day 28 after booster, $n = 10$). All COVAC participants verbally reported CMs and gave consent to review their medical records. Electronic medical record reviews were undertaken for each individual participant and evidence of the following CMs and factors were documented, including previous COVID-19, chronic respiratory disease, renal disease, diabetes, pregnancy, liver disease, returned traveler, cardiac disease, obesity (BMI > 30), neurological disease, nursing home resident, if homeless, immunosuppressed or other conditions and statuses (specified in free text). Details of COVID-19 vaccinated NI participants with CMs (diabetes and renal disease; $n = 22$ individuals) as well as inflammatory bowel syndrome ($n = 31$)[51] are listed in Supplementary Table 2. Details of the DISI cohort ($n = 15$) have been described previously[25,52].

Peripheral blood was collected in heparinized or EDTA tubes and serum tubes, with plasma and sera collected after centrifugation, respectively. Peripheral blood monocular cells (PBMCs) were isolated via Ficoll-Paque separation. Samples were processed at the Menzies School of Health Research or at the University of Melbourne, and stored at the University of Melbourne. Demographic, clinical and sampling information of COVAC, DISI and NI participants with CMs are described in Supplementary Tables 1 and 2. Experiments conformed to the principles of the Declaration of Helsinki (2013) and the Australian National Health and Medical Research Council Code of Practice. Written informed consent was obtained from all blood donors before the study. Participants of the current study were not compensated. The study was approved by the Human Research Ethics Committee of the Northern Territory Department of Health and the Menzies School of Health Research (no. 2012-1928, COVAC, LIFT), The Alfred Hospital (no. 280-14, DISI), the Royal Melbourne Hospital with approval from Melbourne Health (nos. HREC/74403/MH-2021 and HREC/17/MH/53), Central Adelaide Local Health Network Human (CALHN) Research Ethics Committee (no. HREC 14541; recruitment was part of the REVAX trial (ANZCTR ID: ACTRN12621000532808), La Trobe Human Ethics Committee (no. HEC21097), Austin Hospital Ethics Committee (no. HREC/75984/Austin-2021) and the University of Melbourne Human Research Ethics Committees (nos. 11077, 21864, 11124 and 15398-3).

### SARS-CoV-2 RBD ELISAs

Ancestral, Delta and Omicron RBD-specific ELISAs to detect IgG antibodies were performed as described previously[19,25,52]. In brief, Nunc MaxiSorp flat-bottom 96-well plates (Thermo Fisher Scientific) were coated with RBD (2 µg ml$^{-1}$), blocked with PBS (with w/v 1% BSA) and incubated with plasma samples serial-diluted in PBS (with v/v 0.05% Tween 20 and w/v 0.5% BSA). Samples were read on a Multiskan plate reader (Labsystems) using the Thermo Ascent Software for Multiskan v.2.4. Inter- and intra-experimental measurements were normalized using a positive control plasma from an individual with COVID-19. Endpoint titers were determined by interpolation from a sigmoidal curve fit (all $R^2 > 0.95$; Prism 9 (GraphPad Software)) as the reciprocal dilution of plasma that produced more than 15% absorbance of the positive control at 1:31.6. Seroconversion was defined as at least a fourfold increase in 'un-log' antibody titer from V1 plasma. Seropositivity was defined as the mean + 2 × s.d. of the pooled FN and NI V1 RBD IgG titer.

### Microneutralization assay

Microneutralization activity of serum samples was assessed as described previously[25,52,53]. The SARS-CoV-2 isolate CoV/Australia/VIC01/2020 (ref. 54) was propagated in Vero cells and stored at −80 °C. Sera were heat-inactivated at 56 °C for 30 min and serially diluted. Residual virus infectivity in the serum and virus mixtures was assessed in quadruplicate wells of Vero cells incubated in serum-free medium containing 1 µg ml$^{-1}$ of tosyl phenylalanyl chloromethyl ketone trypsin at 37 °C and 5% $CO_2$. The viral cytopathic effect was read on day 5. The neutralizing antibody titer was calculated using the Reed–Muench method.

### Avidity assay

The avidity of RBD-specific IgG was assessed by urea-mediated dissociation ELISA. Nunc-Immuno MaxiSorp flat-bottom 96-well plates (Thermo Fisher Scientific) were coated with RBD protein overnight at 4 °C. Plates were washed and blocked with PBS (with w/v 1% BSA) for at least 1 h. Donor plasma was added in $\log_{0.5}$ dilutions and incubated for 2 h at room temperature. Wells were washed and 6 M urea added and incubated for 15 min. Bound antibodies were then detected using horseradish peroxidase-conjugated anti-human IgG as described previously[19,25,52]. The amount (in percentage) of antibody remaining was determined by comparing the total area of the antibody titration curve (across four dilutions) in the presence and absence of urea treatment and is expressed as the avidity index.

### IgG purification and IgG N-linked glycan profiling

IgG antibodies were purified from plasma as described previously[55]. In brief, total IgG was collected using the Melon Gel IgG Purification Kit (Thermo Fisher Scientific) according to the manufacturer's protocol. Purified IgG samples were then centrifuged through 100-kDa Amicon Ultra Centrifugal Filters (Merck Millipore) at 14,000$g$ for 15 min to remove excess serum proteins and buffer exchange antibodies into PBS. Purity was confirmed via SDS–polyacrylamide gel electrophoresis (Bio-Rad Laboratories) and IgG concentrations were measured using a NanoDrop spectrophotometer (Bio-Rad Laboratories). IgG N-linked glycosylation patterns were measured according to the ProfilerPro glycan profiling LabChip GXII Touch protocol on the LabChip GXII Touch HT Microchip-CE platform (PerkinElmer) using the LabChip GX Touch software (v.1.9.1010.0), as described previously[55]. Microchip capillary electrophoresis laser-induced fluorescence analysis of digested and labeled N-linked glycans was performed. The relative prevalence of major N-linked glycan profiles of IgG was analyzed using the LabChip GX Reviewer (PerkinElmer) v.5.4.2222.0. Peaks were assigned based on the migration of known standards and glycan digests. The peak area and relative prevalence of each glycan pattern were calculated.

### Cytokine analysis

Donor plasma was diluted 1:2 to measure cytokines using the LEGENDplex Human Inflammation Panel 1 kit (BioLegend) according to the manufacturer's instructions. Cytokines and chemokines including IL-1β, IFNα2, IFNγ, TNF, MCP-1 (CCL2), IL-6, IL-8 (CXCL8), IL-10, IL-12p70, IL-17A, IL-18, IL-23 and IL-33 were measured. Samples were acquired on a FACSCanto II cytometer (BD Biosciences) and analyzed with the QOGNIT LEGENDplex program.

### AIM and ICS assays

Thawed PBMCs were plated onto a 96-well plate at $1 × 10^6$ PBMCs per well. In the AIM assay, cells were stimulated in complete Roswell Park Memorial Institute medium with 10 µg ml$^{-1}$ SARS-CoV-2 spike peptide pool (181 peptides, 0.06 µg ml$^{-1}$ per peptide) or dimethylsulfoxide (DMSO) and cultured at 37 °C and 5% $CO_2$ for 24 h. Cells were then washed and stained with a panel of titrated cell-surface markers including CXCR5-BV421 (1:25, catalog no. 562747, BD Biosciences), CD3-BV510 (1:200, catalog no. 317332, BioLegend), CD8-BV605 (1:100, catalog no. 564116, BD Biosciences), CD4-BV650 (1:200, catalog no. 563875, BD Biosciences), CD25-BV711 (1:200, catalog no. 563159, BD Biosciences), CXCR3-BV785 (1:20, catalog no. 353738, BioLegend), CD137-APC (1:50, catalog no. 309810, BioLegend), CD27-AF700 (1:50,

catalog no. 560611, BD Biosciences), CD14/CD19-APC-H7 (1:100, catalog no. 560180/560252, BD Biosciences), LIVE/DEAD near-infrared (NIR) (1:800, catalog no. L34976, Invitrogen), CD69-PerCP-Cy5.5 (1:50, catalog no. 310925, BioLegend), CD134-PE (1:200, catalog no. 340420, BD Biosciences), CD95-PE-CF594 (1:100, catalog no. 562395, BD Biosciences) and CD45RA-PeCy7 (1:50, catalog no. 337167, BD Biosciences) before fixing with 1% paraformaldehyde (PFA). In the ICS assay, cells were stimulated with 100 µg ml$^{-1}$ spike peptide pool or DMSO in combination with anti-CD28/CD49d (1:100, catalog no. 347690, BD Biosciences) and 10 U ml$^{-1}$ IL-2 (catalog no. 11147528001, Roche), with brefeldin A (catalog no. 555029, BD Biosciences) added after 5 h. Cells were stained with CD3-BV510 (1:200), CD4-BV650 (1:200), CD8-PerCPCy5.5 (1:100, catalog no. 565310, BD Biosciences) and LIVE/DEAD NIR (1:800), fixed using the BD Cytofix/Cytoperm kit (catalog no. 554723, BD Biosciences) and stained intracellularly with IFNγ-V450 (1:100, catalog no. 560371, BD Biosciences), MIP-1β-APC (1:40, catalog no. 560656, BD Biosciences) and TNFα-AF700 (1:50, catalog no. 557996, BD Biosciences). Acquisition was on an LSRII Fortessa.

## Peptide-HLA class I and class II tetramers
Tetramers were generated from soluble, biotinylated HLA-DPB1*04:01 S$_{167-180}$ (TFEYVSQPFLMDLE), HLA-A*01:01 S$_{865-873}$ (LTDEMIAQY), HLA-A*02:01 S$_{269-277}$ (YLQPRTFLL), HLA-A*03:01 S$_{378-386}$ (KCYGVSPTK), HLA-A*24:02 S$_{1,208-1,217}$ (QYIKWPWYI) monomers. Briefly, HLA α-heavy chain with a C-terminal BirA biotinylation motif and β2 microglobulin were expressed and purified as inclusion bodies in *Escherichia coli*, solubilized in 6 M guanidine HCl and refolded with corresponding spike peptides in buffer containing 50 mM Tris pH 8, 3 M urea, 0.4 M arginine, 2 mM oxidized glutathione, 20 mM glutathione, 2 mM EDTA, 10 mM phenylmethylsulfonyl fluoride and cOmplete protease inhibitor (Roche). After dialysis in 10 mM Tris, HLA monomers were purified via diethylaminoethyl and HiTrapQ ion exchange chromatography, and biotinylated with BirA ligase in 50 mM bicine pH 8.3, 10 mM ATP, 10 mM magnesium acetate and 100 µm D-biotin. After S200 gel permeation chromatography, fully biotinylated HLA monomers were stored at −80 °C and conjugated to fluorescently labeled streptavidin (SA), phycoerythrin (PE)-SA or allophycocyanin (APC)-SA (BD Biosciences) at an 8:1 monomer to SA molar ratio to form pHLA tetramers.

## Ex vivo tetramer enrichment
PBMCs ($3 \times 10^6$–$15 \times 10^6$) were blocked with FcR block and NKB1 (if using the A24/S$_{1,208}$-APC tetramer) for 15 min on ice, followed by staining with DPB4/S$_{167}$-PE, A1/S$_{865}$-PE or APC, A2/S$_{269}$-PE or APC, A3/S$_{378}$-PE or APC or A24/S$_{1,208}$-APC tetramers at room temperature for 1 h in MACS buffer (PBS with 0.5% BSA and 2 mM EDTA). Cells were then incubated with anti-PE and/or anti-APC microbeads (Miltenyi Biotec) and tetramer$^+$ cells were enriched using magnetic separation[29,33]. Lymphocytes were stained with anti-CD71-BV421 (1:50, catalog no. 562995), anti-CD4-BV650 (1:200), anti-CD27-BV711 (1:200, catalog no. 563167), anti-CD38-BV786 (1:100, catalog no. 563964), anti-CCR7-AF700 (1:25, catalog no. 561143), anti-CD14-APC-H7 (1:100), anti-CD19-APC-H7 (1:100, catalog no. 560177), anti-CD45RA-FITC (1:200, catalog no. 555488), anti-CD8-PerCP-Cy5.5 (1:50), anti-CD95-PE-CF594 (1:100), anti-PD1-PE-Cy7 (1:50, catalog no. 561272) (BD Biosciences), anti-CD3-BV510 (1:200), anti-HLA-DR-BV605 (1:100, catalog no. 307640) (BioLegend) and LIVE/DEAD NIR (1:800, catalog no. L10119, Invitrogen) stain for 30 min, washed, resuspended in MACS buffer and analyzed using flow cytometry. In selected experiments, cells were fixed with 1% PFA and washed before being acquired on an LSRII Fortessa or single-cell-sorted using the FACSAria III (BD Biosciences) for the TCR analyses. FCS files were analyzed using FlowJo v.10. Samples with cell counts of tetramer$^+$CD4$^+$ or tetramer$^+$CD8$^+$ T cells below ten were not characterized further phenotypically using the staining panel. Samples with 0 Tet$^+$ T cells were shown on the *x* axis for visualization.

## TCRαβ repertoire analysis
Tetramer$^+$CD4$^+$ and CD8$^+$ T cells were single-cell-sorted into empty 96-well twin.tec PCR plates (Eppendorf), centrifuged then stored at −80 °C. Multiplex-nested PCR with reverse transcription amplification of paired CDR3α and CDR3β regions was performed as described elsewhere[56,57] using the primers listed in Supplementary Table 6. TCR sequences were analyzed using IMGT/V-QUEST. Alluvial plots were generated in R v.4.2.1 using ggalluvial v.0.12.3 (ref. 58) (http://corybrunson.github.io/ggalluvial/) and formatted in Inkscape v.1.x. The Simpson diversity index (*D*) for TCR diversity was calculated as $D = 1 - \frac{\sum n(n-1)}{N(N-1)}$, where *n* is the number sequence of a given clonotype and *N* is the total number of sequences within the group. TCR sequences were parsed and annotated using TCRdist[59] v.0.0.2 for downstream analysis, with nucleotide call qualities ignored due to manual base calling. Resulting TCRαβ clonotypes were analyzed for differences in segment use by first considering *TRAV* segment frequencies across FN and NI adults for the A2/S$_{269}$ epitope. The cord diagrams were made using TCRdist3 (refs. 60) v.0.2.2. To investigate allelic variation in germline TCR V or J regions, the reference (GenBank ID: AE000659.1) from the IMGT *TRAV12-1*01*, *TRAV12-2*01*, *TRAV12-3*01* alleles was built using bowtie2 v.2.5.0; then all the reads were aligned to the reference. To investigate potential differences in noncoding regions leading to TCR gene segment use, the TCRα repertoire was reconstructed using mixcr v.3.0.13 to estimate V use of *TRAV12-1*.

## Assessment of SARS-CoV-2 spike-specific B cells
Spike-specific B cell responses toward the vaccine strain were assessed using thawed PBMCs or TAME-flow through fractions. Cells were stained with spike recombinant probes conjugated to PE, fixed and acquired on an LSRII Fortessa, as described elsewhere[29,61]. Samples with 0 spike-specific B cells were shown on the *x* axis for visualization. Four samples were excluded from the spike-specific B cell analyses due to minimal numbers of lymphocytes or CD19$^+$ B cells.

## Statistical analyses
No statistical methods were used to predetermine sample sizes but our sample sizes are similar to those reported in a previous publication[51]. Normality tests were not performed and nonparametric statistical analyses were performed in the study. Statistical significance was assessed using a two-tailed Mann–Whitney *U*-test, two-tailed Wilcoxon signed-rank test, two-way ANOVA and Spearman correlation coefficient ($r_s$) in Prism 9 unless stated otherwise. For the correlation matrices and volcano plots, cytokine concentrations and microneutralization values were log$_{10}$-transformed before analysis. Pearson *r* was calculated in R (v.4.2.1)[62] (https://www.R-project.org/) using psych v.2.2.5 (ref. 63) (https://CRAN.R-project.org/package=psych) with false discovery rate (FDR) adjustment of *P* values. Student's *t*-tests for the volcano plots were calculated using rstatix v.0.7.0 (ref. 64) (https://CRAN.R-project.org/package=rstatix), with FDR adjustment for multiple comparisons. Unless otherwise stated, an FDR < 0.05 was considered statistically significant. Correlation matrices were prepared in R using corrplot v.0.92 (ref. 65) (https://github.com/taiyun/corrplot). Volcano plots were created using EnhancedVolcano v.1.14.0 (ref. 66) (https://github.com/kevinblighe/EnhancedVolcano). Multiple linear regression comparing the contributions of demographic and immunological factors to (log-transformed) V3 RBD IgG responses were performed in R v.4.2.0 using the LM function of the nlme package v.3.1-160. Only data on Pfizer-vaccinated individuals sampled within 45 days of vaccination were included in this regression.

## Reporting summary
Further information on research design is available in the Nature Portfolio Reporting Summary linked to this article.

## Data availability

The published article includes all datasets generated or analyzed during the study. The TCR sequences in this study were uploaded to Mendeley Data under the following access code: https://doi.org/10.17632/fj636xh5y6.1. Raw FACS data are shown in the manuscript. FACS source files are available from the authors upon reasonable request. Source data are provided with this paper.

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

## Acknowledgements

Aboriginal and Torres Strait Islanders in Australia were respectfully referred to as Australian FN peoples and FN in this paper, as dictated by space and word limit. We thank P. Binks, K. Hosking, T. De Santis, C. Marshall, B. Patel, E. Gargan, J. Webb, M. Dickson, L. H. E. Vintour-Cesar, M. McKinnon, C. Van Wessel and C. Woerle from the Menzies School of Health research and V. Baghbanian, S. Moore and L. Thomas from the Central Australia Aboriginal Congress for participant recruitment and blood processing. This work was supported by the National Health and Medical Research Council (NHMRC) Leadership Investigator Grant to K.K. (no. 1173871) and A.C.C. (no. 1194678), Research Grants Council of the Hong Kong Special Administrative Region (no. T11-712/19-N) to K.K., a Medical Research Future Fund (MRFF) award no. 2005544 to K.K., S.J.K., A.W.C., A.C.C. and J.D., a MRFF award no. 2016062 to K.K., T.H.O.N., L.C.R., S.J.K., J.D., A.W.C., A.C.C., M.P.D., P.G.T., D.S.K. and J.R., an NHMRC Emerging Leadership Level 1 Investigator Grant to T.H.O.N. (no. 1194036), an NHMRC Emerging Leadership Level 2 Investigator Grant to A.W.C. (no. 2009092) and a 2021 Doherty Agility Fund to T.H.O.N., K.K., K.S., A.W.C. and W.Z. W.Z. is supported by the Melbourne Research Scholarship from the University of Melbourne. S.J.K. is supported by an NHMRC Senior Principal Research Fellowship (no. 1136322). J.R. is supported by an Australian Research Council Laureate Fellowship. J.C.C., A.A.M., M.V.P. and P.G.T. are supported by National Institutes of Health/National Institute of Allergy and Infectious Diseases (NIH/NIAID) grant nos. R01AI136514 and U01AI150747, and by the American Lebanese Syrian Associated Charities at St. Jude Children's Research Hospital. S.G. is supported by an NHMRC Senior Research Fellowship (no. 1159272) and the MRFF. This project has been funded in whole or in part with Federal funds from the NIAID/NIH, Department of Health and Human Services, under contract no. 75N93021C00018 (NIAID Centers of Excellence for Influenza Research and Response, CEIRR). This work was supported by the Australian Government Commonwealth Contract-Services (no. E21-24684) to facilitate Central Australia involvement. We thank the Melbourne Cytometry Platform (Peter Doherty Institute and Melbourne Brain Centre nodes) for provision of flow cytometry services. We thank BEI Resources, NIAID and NIH for providing the peptide array, SARS-related coronavirus 2 spike glycoprotein (NR-52402). This research included samples and data from the Sentinel Travelers Research Preparedness Platform for Emerging Infectious Diseases (SETREP-ID). We thank all SETREP-ID investigators and sites and all participants involved. We thank B. Scher for setting up the ethics and governance for the SETREP-ID platform and the Australian Partnership for Preparedness Research on Infectious Disease Emergencies (APPRISE) for ongoing funding of SETREP-ID. We thank A. Rhodes, J. Chang, A. Dantanarayana and R. Cao who contributed to the SETREP-ID biobank. SETREP-ID is supported by funding from the NHMRC Centre of Research Excellence, APPRISE (ID 1116530), the Snow Medical Foundation, the Jack Ma Foundation and the A2 Milk Company. This research was conducted using samples and data from the Victorian Critical Vaccinees Collection (VC2). We acknowledge the VC2 investigators and sites and the support of the Victorian Government, and thank all the participants involved.

## Author contributions

K.K. led the study. K.K., J.D. and T.H.O.N. supervised the study. K.K., T.H.O.N., W.Z., L.K., B.Y.C., L.C.R., L.F.A., K.S. and A.W.C. designed the experiments. W.Z., L.K., B.Y.C., L.C.R., L.F.A., R.A.P., H.-X.T., E.B.C. and F.M. performed and analyzed the experiments. G.B.P., B.G.-B., S.G., J.A.J., J.A., M.M. and V.R. processed and shipped the samples. H.A.M., K.S., A.A.M., M.V.P., J.C.C., D.S.K., M.P.D., P.G.T. and A.W.C. analyzed the data. J.P., P.C., K.S., F.K., A.K.W., S.J.K. and J.R. provided crucial reagents. C.L., B.F.M., J.B., S.L., J.N. and J.D. recruited the COVAC and LIFT COVID-19 cohorts. A.C.C. recruited the DISI cohort. S.G., P.T.C., I.T., N.E.H., B.C., E.Z. and S.J.K. recruited the NI cohorts. A.M. provided intellectual input for conducting the FN Indigenous Australian research. K.K., J.D., T.H.O.N. and W.Z. wrote the manuscript. All authors reviewed and approved the manuscript.

## Competing interests

The Icahn School of Medicine at Mount Sinai has filed patent applications relating to SARS-CoV-2 serological assays and Newcastle

disease virus-based SARS-CoV-2 vaccines that list F.K. as coinventor. Mount Sinai has spun out a company, Kantaro, to market serological tests for SARS-CoV-2. F.K. has consulted for Merck and Pfizer (before 2020), and is currently consulting for Pfizer, CSL Seqirus and Avimex. The Krammer laboratory is also collaborating with Pfizer on animal models of SARS-CoV-2. H.A.M. is currently consulting for ENA Respiratory. P.T. is on the scientific advisory board of ImmunoScape and CytoAgents and has consulted for Johnson & Johnson. P.T. has received travel support and honoraria from Illumina and 10X Genomics and has patents related to TCR discovery and expression. The other authors declare no competing interests.

## Additional information

**Extended data** is available for this paper at https://doi.org/10.1038/s41590-023-01508-y.

**Correspondence and requests for materials** should be addressed to Thi H. O. Nguyen, Jane Davies or Katherine Kedzierska.

**Peer review information** *Nature Immunology* thanks Luis Graca, Paul Moss, and the other, anonymous, reviewer(s) for their contribution to the peer review of this manuscript. Primary Handling Editor: Ioana Visan was the primary editor on this article and managed its editorial process and peer review in collaboration with the rest of the *Nature Immunology* editorial team. Peer reviewer reports are available.

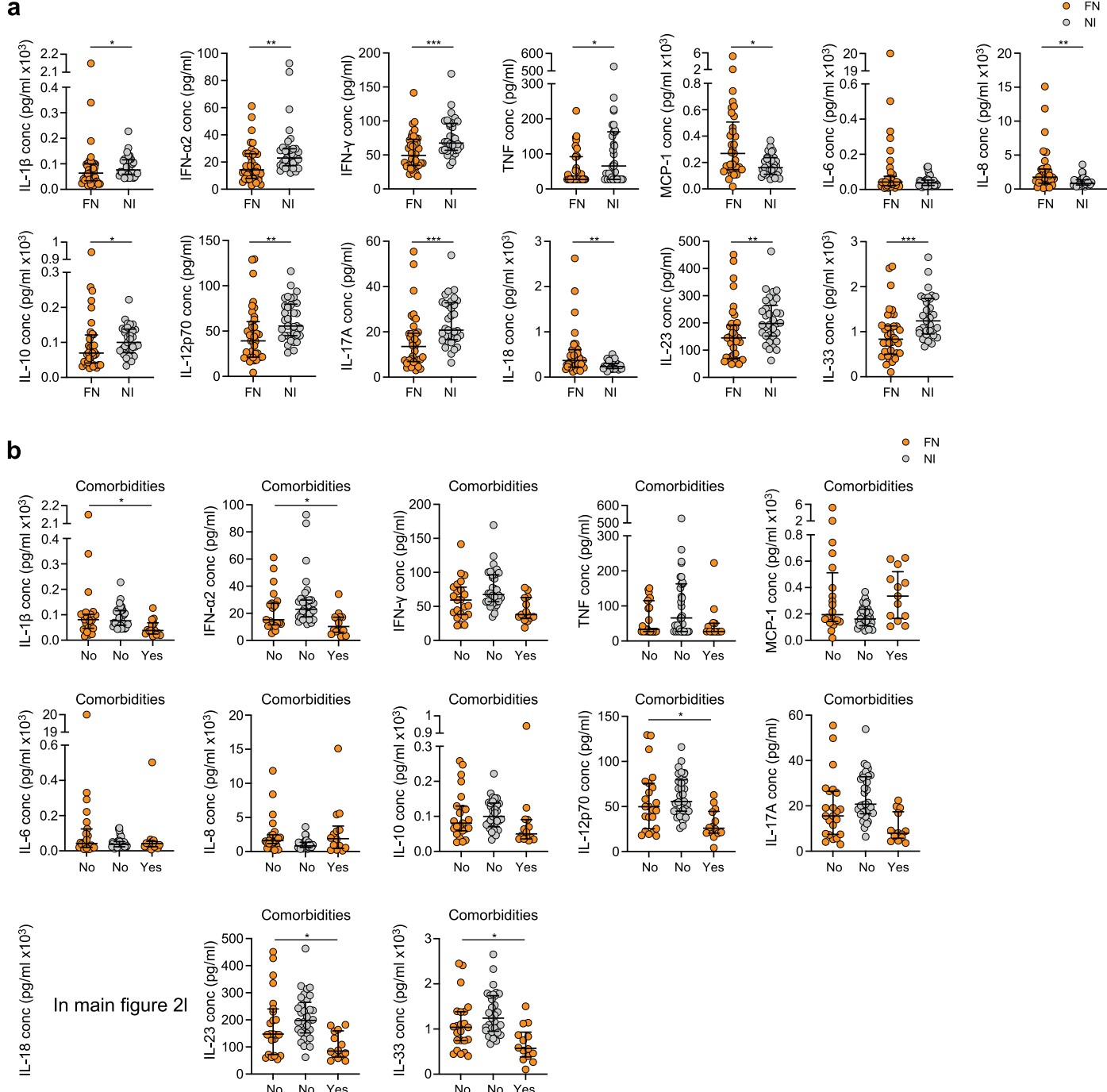

**Extended Data Fig. 1 | Concentration of 13 cytokines and chemokines in First Nations and non-Indigenous individuals. a, b,** V1 concentration of 13 cytokines and chemokines in First Nations (FN) and non-Indigenous (NI) individuals (**a**; $P_{IL-1\beta}$ = 0.0234, $P_{IFN-\alpha2}$ = 0.0010, $P_{IFN-\gamma}$ = 0.0008, $P_{TNF}$ = 0.0199, $P_{MCP-1}$ = 0.0108, $P_{IL-8}$ = 0.0041, $P_{IL-10}$ = 0.0392, $P_{IL-12p70}$ = 0.0025, $P_{IL-17A}$ = 0.0003, $P_{IL-18}$ = 0.0022, $P_{IL-23}$ = 0.0022, $P_{IL-33}$ = 0.0008) and individuals with or without comorbidities

(**b**; FN No comorbidities vs Comorbidities: $P_{IL-1\beta}$ = 0.0162, $P_{IFN-\alpha2}$ = 0.0398, $P_{IL-12p70}$ = 0.0163, $P_{IL-23}$ = 0.0353, $P_{IL-33}$ = 0.0367). Bars indicate median with interquartile range. Statistical significance was determined with two-sided Mann-Whitney test. $n_{FN}$ = 37 ($n_{No\ comorbidities}$ = 23, $n_{Comorbidities}$ = 14), $n_{NI}$ = 33. Please refer to Source Data Fig. 2k for Source Data.

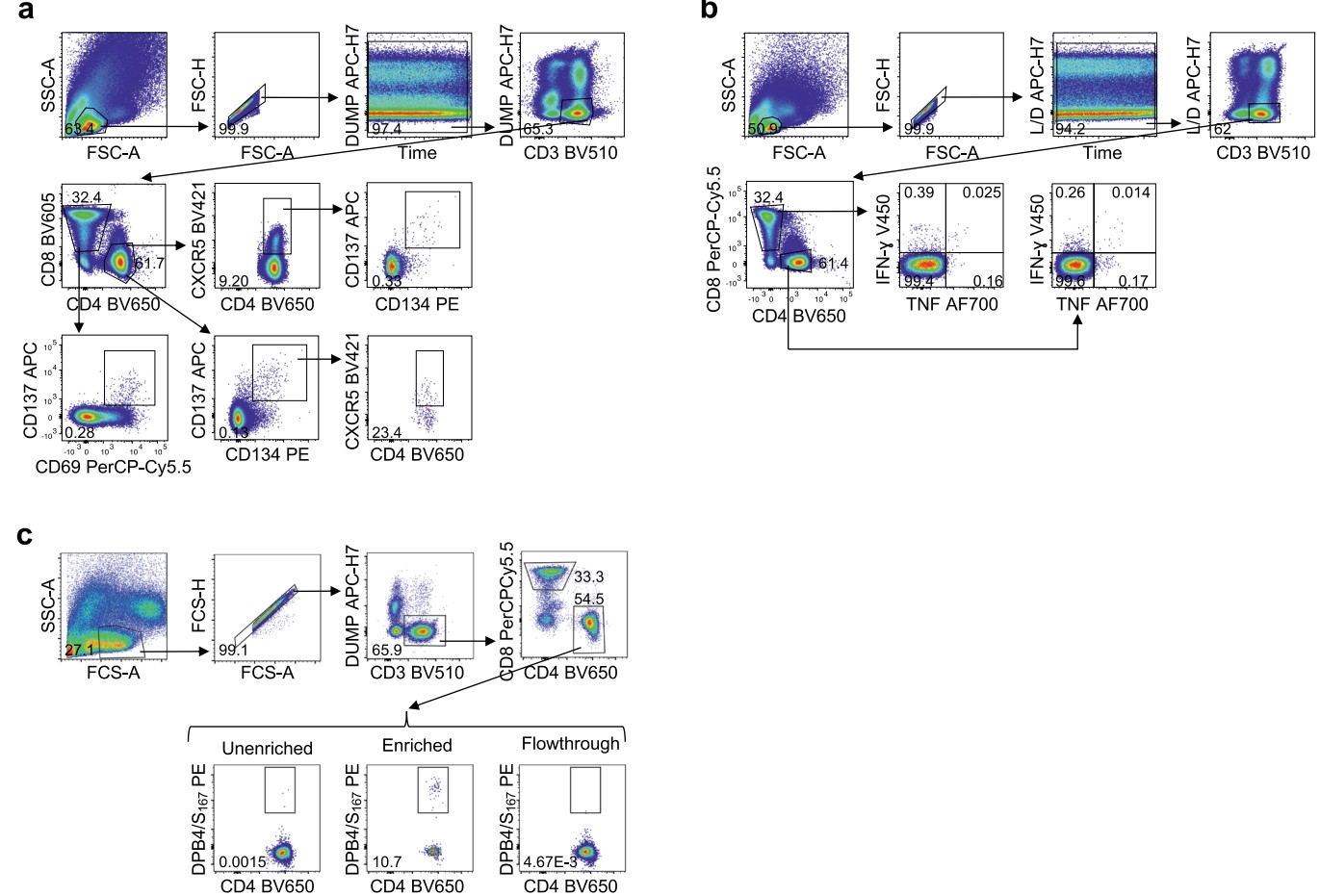

**Extended Data Fig. 2 | Gating strategies of flow cytometry assays. a–c**, Gating strategy of Activation-Induced Markers (**a**), Intracellular Cytokine Staining (**b**) and Tetramer enrichment (**c**) assays.

## CDR3α

## CDR3β

**Extended Data Fig. 3 | Detailed figure legends for TCR CDR3 regions.**

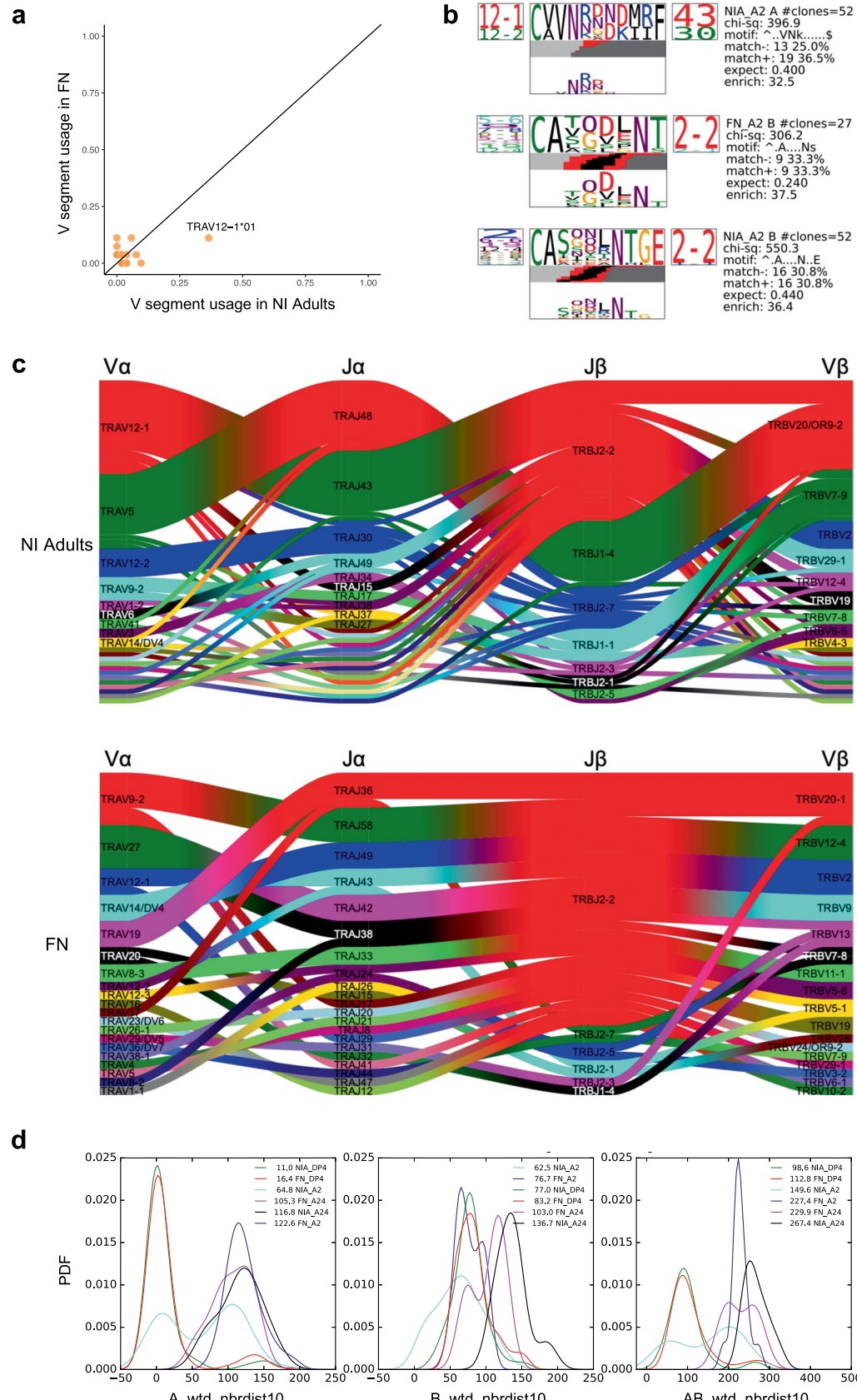

**Extended Data Fig. 4 | See next page for caption.**

**Extended Data Fig. 4 | Statistical analyses of TCR gene segment usage in First Nations and non-Indigenous individuals. a**, Scatter plot of frequency of TRAV segment in A2/$S_{269}$-specific clonotypes by population. **b**, A subset of statistically significant motifs detected for the A2/$S_{269}$ response. **c**, Cord plots depicting v-segment and j-segment pairings across TCR clonotypes for the A2/$S_{269}$ response for NIA and FN populations. **d**, Smoothed frequency distributions of averaged nearest-neighbors TCRdist scores across NIA and FN populations for distinct epitopes. Left: alpha chains; middle: beta chains; right: paired alpha-beta chains. Numerical values in the legend correspond to the average neighbor distance for that repertoire. Lower values on the x-axis represent more similarity across TCR clonotypes. NIA, Non-Indigenous Adults; FN, First Nations Adults. 'A' indicates alpha chain, 'B' indicates beta chain. See Dash et al.[59] for additional information on TCRdist and resulting characterizations.

# Reporting Summary

## Statistics

For all statistical analyses, confirm that the following items are present in the figure legend, table legend, main text, or Methods section.

| n/a | Confirmed | |
|---|---|---|
| ☐ | ☒ | The exact sample size (*n*) for each experimental group/condition, given as a discrete number and unit of measurement |
| ☐ | ☒ | A statement on whether measurements were taken from distinct samples or whether the same sample was measured repeatedly |
| ☐ | ☒ | The statistical test(s) used AND whether they are one- or two-sided<br>*Only common tests should be described solely by name; describe more complex techniques in the Methods section.* |
| ☐ | ☒ | A description of all covariates tested |
| ☐ | ☒ | A description of any assumptions or corrections, such as tests of normality and adjustment for multiple comparisons |
| ☐ | ☒ | A full description of the statistical parameters including central tendency (e.g. means) or other basic estimates (e.g. regression coefficient) AND variation (e.g. standard deviation) or associated estimates of uncertainty (e.g. confidence intervals) |
| ☐ | ☒ | For null hypothesis testing, the test statistic (e.g. *F*, *t*, *r*) with confidence intervals, effect sizes, degrees of freedom and *P* value noted<br>*Give P values as exact values whenever suitable.* |
| ☒ | ☐ | For Bayesian analysis, information on the choice of priors and Markov chain Monte Carlo settings |
| ☒ | ☐ | For hierarchical and complex designs, identification of the appropriate level for tests and full reporting of outcomes |
| ☐ | ☒ | Estimates of effect sizes (e.g. Cohen's *d*, Pearson's *r*), indicating how they were calculated |

*Our web collection on statistics for biologists contains articles on many of the points above.*

## Software and code

Policy information about availability of computer code

| Data collection | BD FACS DIVA v8.0.1; Thermo Ascent Software for Multiskan v2.4; LabChip GXII Touch HT Microchip-CE platform |
|---|---|
| Data analysis | FlowJo v10; Prism v9; R v4.2.0, v4.2.1, ggalluvial v0.12.3, Inkscape v1.2, psych v2.2.5, rstatix v0.7.0, corrplot v0.92, EnhancedVolcano v1.14.0; online QOGNIT LEGENDplex™ program; LabChip GX Touch software v1.9.1010.0; LabChip GX Reviewer software v5.4.2222.0, TCRdist v0.0.2, TCRdist3 v0.2.2, R package NLME v3.1-160, bowtie2 program v2.5.0, mixcr v3.0.13 |

For manuscripts utilizing custom algorithms or software that are central to the research but not yet described in published literature, software must be made available to editors and reviewers. We strongly encourage code deposition in a community repository (e.g. GitHub). See the Nature Portfolio guidelines for submitting code & software for further information.

## Data

Policy information about availability of data

All manuscripts must include a data availability statement. This statement should provide the following information, where applicable:
- Accession codes, unique identifiers, or web links for publicly available datasets
- A description of any restrictions on data availability
- For clinical datasets or third party data, please ensure that the statement adheres to our policy

The published article includes all datasets generated or analyzed during the study. Source data are provided with this paper as Source Date files. TCR sequences in this study were uploaded to Mendeley Data, with the access code DOI: 10.17632/fj636xh5y6.1. Raw FACS data are shown in the manuscript. FACS-source files are available from the authors upon request. The majority data of the current study are based on flow cytometry and there are hundreds of FACS-source files with a big file size. We are happy to provide any single file upon request.

# Field-specific reporting

Please select the one below that is the best fit for your research. If you are not sure, read the appropriate sections before making your selection.

☒ Life sciences        ☐ Behavioural & social sciences        ☐ Ecological, evolutionary & environmental sciences

For a reference copy of the document with all sections, see nature.com/documents/nr-reporting-summary-flat.pdf

# Life sciences study design

All studies must disclose on these points even when the disclosure is negative.

| | |
|---|---|
| Sample size | The sample size was determined by the availability of samples from Australian First Nations and non-Indigenous participants that received the BNT162b2 COVID-19 vaccine in Northern Territory, Australia in 2021-2022. The sample size of non-Indigenous participants with comorbidities (renal disease, diabetes, IBD) was determined by the availability of samples, with recruitment performed through larger cohorts. |
| Data exclusions | No data were excluded with the following exception which was pre-established: donors who had a total number of less than 10 counted tetramer+CD8+ or tetramer+CD4+ T cells within the whole enriched fraction were excluded for further phenotypic analyses as cell numbers were too low (Fig 5h). This was indicated in the manuscript in Methods. 4 samples were excluded from spike-specific B cell analyses due to minimal numbers of lymphocytes or CD19+ B cells. A source data file was generated to show all data points. |
| Replication | Experiments could not be replicated due to limited PBMC numbers. These are rare and unique patient samples and so we were limited to performing all the available assays. To ensure reliability, all timepoints from the same patient were carried out in the same experiment. |
| Randomization | Randomization was not applicable to the study, participants received BNT162b2 or ChAdOx1-S COVID-19 vaccines following the current vaccine recommendations in Australia. |
| Blinding | Experiments were not blinded as specific experiments were designed for COVID-19 vaccinees, for example HLA type was needed to perform tetramer studies. |

# Reporting for specific materials, systems and methods

We require information from authors about some types of materials, experimental systems and methods used in many studies. Here, indicate whether each material, system or method listed is relevant to your study. If you are not sure if a list item applies to your research, read the appropriate section before selecting a response.

## Materials & experimental systems

| n/a | Involved in the study |
|---|---|
| ☐ | ☒ Antibodies |
| ☐ | ☒ Eukaryotic cell lines |
| ☒ | ☐ Palaeontology and archaeology |
| ☒ | ☐ Animals and other organisms |
| ☐ | ☒ Human research participants |
| ☒ | ☐ Clinical data |
| ☒ | ☐ Dual use research of concern |

## Methods

| n/a | Involved in the study |
|---|---|
| ☒ | ☐ ChIP-seq |
| ☐ | ☒ Flow cytometry |
| ☒ | ☐ MRI-based neuroimaging |

# Antibodies

| | |
|---|---|
| Antibodies used | We used commercially-available antibodies as per Material and Methods.<br>AIM assay: anti-CXCR5-BV421 (562747; BD Biosciences; clone RF8B2), anti-CD3-BV510 (317332; BioLegend; clone OKT3), anti-CD8-BV605 (564116; BD Biosciences; clone SK1), anti-CD4-BV650 (563875; BD Biosciences; clone SK3), anti-CD25-BV711 (563159; BD Biosciences; clone 2A3), anti-CXCR3-BV785 (353738; BioLegend; clone G025H7), anti-CD137-APC (309810; BioLegend; clone 4B4-1), anti-CD27-AF700 (560611; BD Biosciences; clone M-T271), anti-CD14/CD19-APC-H7 (560180 clone MφP9/560252 clone SJ25C1; BD Biosciences), anti-CD69-PerCPCy5.5 (310925; BioLegend; clone FN50), anti-CD134-PE (340420; BD Biosciences; clone L106), anti-CD95-PE-CF594 (562395; BD Biosciences; clone DX2), anti-CD45RA-PeCy7 (337167; BD Biosciences; clone L48).<br>ICS assay: anti-CD3-BV510 (317332; BioLegend; clone OKT3), anti-CD4-BV650 (563875; BD Biosciences; clone SK3), anti-CD8-PerCPCy5.5 (565310; BD Biosciences; clone SK1) anti-IFNg-v450 (560371; BD Biosciences; clone B27), anti-MIP-1b-APC (560656; BD Biosciences; clone D21-1351) anti-TNF-AF700 (557996; BD Biosciences; clone MAb11).<br>Tetramer enrichment assay: anti-CD71-BV421 (#562995; BD Biosciences; clone M-A712), anti-CD4-BV650 (#563875; BD Biosciences; clone SK3), anti-CD27-BV711 (#563167; BD Biosciences; clone L128), anti-CD38-BV786 (#563964; BD Biosciences; clone HIT2), anti-CCR7-AF700 (#561143; BD Biosciences; clone 150503), anti-CD14-APC-H7 (#560180; BD Biosciences; clone MφP9), anti-CD19-APC-H7 (#560177; BD Biosciences; clone SJ25C1), anti-CD45RA-FITC (#555488; BD Biosciences; clone HI100), anti-CD8-PerCP-Cy5.5 (#565310; BD Biosciences; clone SK1), anti-CD95-PE-CF594 (#562395; BD Biosciences; clone DX2), anti-PD1-PE-Cy7 (#561272; BD Biosciences; clone EH12.1), anti-CD3-BV510 (#317332; Biolegend; clone OKT3), anti-HLA-DR-BV605 (#307640; Biolegend; clone L243) |

| Validation | All antibodies were obtained from commercial vendors. Each antibody used had a validated technical data sheet as per manufacturer's website showing positive staining, and titrated in our laboratory to define the appropriate concentration prior to their use. ELISA assay were tested with samples with known high responses from previous assays. FACS positive staining is shown in the FACS plots in the main figures. |
|---|---|

## Eukaryotic cell lines

Policy information about cell lines

| Cell line source(s) | Vero cells were obtained from ATCC (#CCL-81). |
|---|---|
| Authentication | The cell line was not authenticated. |
| Mycoplasma contamination | Vero cells tested mycoplasma negative. |
| Commonly misidentified lines (See ICLAC register) | No commonly misidentified cell lines were used in the study. |

## Human research participants

Policy information about studies involving human research participants

| Population characteristics | Please refer to Supplementary Tables 1-4 for details. |
|---|---|
| Recruitment | Samples were recruited through the Menzies School of Health research in Darwin, Northern Territory; Alfred Hospital; Royal Melbourne Hospital; Austin Hospital; Adelaide Health Network; La Trobe University and University of Melbourne, Australia. Participants were recruited from a wide range of settings including but not limited to: healthcare and university staff vaccination programs, land council community events, remote community vaccination days, hospital and health services campus including a community based dialysis unit. Overall presence of co-morbidities was consistent with reported rates for First Nations people in the Northern Territory. Signed informed consents were obtained from all blood donors prior to the study. |
| Ethics oversight | Experiments conformed to the Declaration of Helsinki Principles and the Australian National Health and Medical Research Council Code of Practice. Written informed consent was obtained from all blood donors prior to the study. The study was approved by the the Human Research Ethics Committee of the Northern Territory Department of Health and Menzies School of Health Research (#2012-1928, COVAC, LIFT), The Alfred Hospital (#280-14, DISI), the Royal Melbourne Hospital (Melbourne, Australia) with approval from Melbourne Health (HREC/74403/MH-2021 and HREC/17/MH/53), Central Adelaide Local Health Network Human Research Ethics Committee; CALHN (HREC 14541; recruitment was part of the REVAX trial (ACTRN12621000532808), La Trobe Human Ethics Committee (HEC21097), Austin Hospital Ethics Committee (HREC/75984/Austin-2021) and the University of Melbourne Human Research Ethics Committees (#11077, #21864 and #11124, #15398-3). |

Note that full information on the approval of the study protocol must also be provided in the manuscript.

## Flow Cytometry

### Plots

Confirm that:

☒ The axis labels state the marker and fluorochrome used (e.g. CD4-FITC).

☒ The axis scales are clearly visible. Include numbers along axes only for bottom left plot of group (a 'group' is an analysis of identical markers).

☒ All plots are contour plots with outliers or pseudocolor plots.

☒ A numerical value for number of cells or percentage (with statistics) is provided.

### Methodology

| Sample preparation | Samples were prepared as described in Methods. Peripheral blood was collected in heparinised or EDTA tubes and serum tubes, with plasma and sera collected after centrifugation, respectively. Peripheral blood monocular cells (PBMCs) were isolated via Ficoll-Paque separation. |
|---|---|
| Instrument | BD LSRII Fortessa, BD FACSAriaIII or BD FACSCanto II was used for acquisition of data |
| Software | BD FACS Diva v8.0.1, FlowJo v10 |
| Cell population abundance | Only single cell sorting was performed, which was confirmed by the presence of single TCR chains. |
| Gating strategy | Gating strategy has been described in the figures, figure legends and Extended Data Figure 2. The Activation-Induced Markers Assay starts with Lymphocyte gate based on FSC-A and SSC-A, then Singlet gate by FSC-A and FSC-H, then a Time gate to ensure signal consistency. Live T cells were gated on CD3+DUMP- cells (DUMP composed of Live |

or Dead, CD14, CD19). CD4+ and CD8+ T cells were gated on CD4+CD8- or CD4-CD8+ cells respectively, followed by gating on CD134+CD137+ cells or CD69+CD137+ cells respectively. CXCR5+CD4+ T follicular helper (TFH), CXCR5+CXCR3+ TFH1, CXCR5 +CXCR3- TFH2 or 17, CXCR5-CD4+ T helper (TH), CXCR5-CXCR3+ TH1, CXCR5-CXCR3- TH2 or 17 cells were also gated for CD134+CD137+ expression.

The Intracellular Cytokine Staining Assay starts with Lymphocyte gate based on FSC-A and SSC-A, then Singlet gate by FSC-A and FSC-H, then a Time gate to ensure signal consistency. Live T cells were gated on Live or Dead-CD3+ cells. CD4+ and CD8+ T cells were gated on CD4+CD8- or CD4-CD8+ cells respectively, followed by gating on IFN-γ+TNF+ cells.

The Tetramer enrichment Assay starts with Lymphocyte gate based on FSC-A and SSC-A, then Singlet gate by FSC-A and FSC-H. Live T cells were gated on CD3+DUMP- cells (DUMP composed of Live or Dead, CD14, CD19). CD4+ and CD8+ T cells were gated on CD4+CD8- or CD4-CD8+ cells respectively. CD4+ and CD8+ T cells were then gated on tetramer staining, followed by further phenotyping including CD27+CD45RA+CCR7+CD95- Naive T cells, CD27+CD45RA+CD95+ TSCM cells, CD27+CD45RA- TCM-like cells, CD27-CD45RA- TEM-like cells, CD27-CD45RA+ TEMRA cells.

☒ Tick this box to confirm that a figure exemplifying the gating strategy is provided in the Supplementary Information.

