## [Peer Review File · Nature Immunology]

Peer Review Information

Journal: Nature Immunology

Manuscript Title: Robust and prototypical immune responses towards COVID-19 vaccine in First Nations people are impacted by comorbidities

Corresponding author name(s): Professor Katherine Kedzierska, Dr Jane Davies, Dr Thi Nguyen

Reviewer Comments & Decisions:

Decision Letter, initial version:
--

11th Nov 2022

Dear Dr. Kedzierska,

Your Article, "Robust and prototypical immune responses towards COVID-19 BNT162b2 vaccine in First Nations people are impacted by comorbidities" has now been seen by 3 referees. While we find your work of considerable potential interest, the reviewers have raised concerns that must be addressed. As such, we cannot accept the current version of the manuscript for publication, but would be happy to consider a revised version that addresses these concerns.

Please revise the manuscript to address all issues raised by the referees. We consider it is important to address the relationship between glycosylation patterns, comorbidities and ethnicity, as pointed out by all referees. At resubmission, please include a point-by-point "Response to referees" detailing how you have addressed each referee comment (please specify page and figure number where the new data can be found in the revised manuscript). This response will be sent back to the referees along with the revised manuscript.

In addition, please include a revised version of any required reporting checklist. It will be available to referees (and, potentially, statisticians) to aid in their evaluation if the manuscript goes back for peer review. A revised checklist is essential for re-review of the paper.

The Reporting Summary can be found here:

When submitting the revised version of your manuscript, please pay close attention to our [href="https://www.nature.com/nature-portfolio/editorial-policies/image-integrity">Digital Image Integrity Guidelines. and to the following points below:](https://www.nature.com/nature-portfolio/editorial-policies/image-integrity)

- that unprocessed scans are clearly labelled and match the gels and western blots presented in figures.
- that control panels for gels and western blots are appropriately described as loading on sample

processing controls

-- all images in the paper are checked for duplication of panels and for splicing of gel lanes.

[REDACTED]

We hope to receive the revised manuscript within 4 months. If you cannot send it within this time, please let us know. We will be happy to consider your revision so long as nothing similar has been accepted for publication at Nature Immunology or published elsewhere.

Nature Immunology is committed to improving transparency in authorship. As part of our efforts in this direction, we are now requesting that all authors identified as 'corresponding author' on published papers create and link their Open Researcher and Contributor Identifier (ORCID) with their account on the Manuscript Tracking System (MTS), prior to acceptance. ORCID helps the scientific community achieve unambiguous attribution of all scholarly contributions. You can create and link your ORCID from the home page of the MTS by clicking on 'Modify my Springer Nature account'. For more information please visit www.springernature.com/orcid.

Thank you for the opportunity to review your work.

Sincerely,

Ioana Visan, Ph.D.
Senior Editor
Nature Immunology

Tel: 212-726-9207
Fax: 212-696-9752
www.nature.com/ni

Reviewers' Comments:

Reviewer #1:

Remarks to the Author:

This is an interesting report that assesses the immune response to vaccination in a cohort of First

Nation donors in Australia.

There is also assessment of the immune response to COVID-19 in hospital in a smaller group

The work is important at this group has been at higher risk of COVID-19 over the last 2 years.

The work seeks to uncover the determinants that might underlie this risk.

The experimental work is very well done and the manuscript is very well written.

It is one of the most comprehensive studies of covid immunity in a cohort that I have seen.

The work is original and novel.

The major finding is that First Nation (FN) have higher levels of co-morbidity. These correlate with higher levels of non-glycosylated antibody and impaired humoral immunity.

As 'co-morbidity' is such a key point in the article it would be important to know how this was assessed. It is stated as 'reported' by the subjects. How was this done, to what degree of granularity and was it objectively assessed. Could there be selective over-reporting of this phenotype in the FN group ?

Perhaps the key point is whether or not the association of co-morbidity with altered 'antibody axis' is seen only in FN subjects or is true for all populations. As I understand it, there were not enough non-FN people with co-morbidity to definitively assess this. This needs further discussion to address.

The work on T cell responses is thorough. It was not entirely clear to me if the responses in the FN group could be because that population has a different profile of HLA alleles. I imagine these were not sequenced ? This is discussed in the paper but could be made clearer.

The work on the hospitalised patients is very thorough but, sadly, I don't really know how it informs the reader. It could be considered for removal.

The team is very comprehensive and talented. One of their talents is in TCR sequence analysis of epitope-specific responses. The emerging data to suggest that there is a potentially different profile of TCR using in the FN subjects is novel. If statistically confirmed this could be an important determinant of the functional quality of the T cell response. What is the statistical assessment of these findings ? The authors may wish to allude as to the degree to which this effect is genetic or acquired. Is there allelic variation in germline TCR V or J regions in FN ? I assume not, and that this represents a selective difference during clonal evolution.

The article is very long and the figures are extensive. I am surprised that this length of article could be considered for print publication. There is a case for rationalising the text and figures.

Reviewer #2:

Remarks to the Author:

The manuscript by Zhang and colleagues describes the immune responses in Australian First Nations people towards vaccination with the COVID-19 BNT162b2 and following COVID-19 infection in non-exposed individuals. This report is of great significance since COVID-19 affected indigenous

populations in very different geographies with particular severity. Yet, despite the disease burden, these populations' immune responses have remained mostly unaddressed. The study provides a comprehensive overview of immune responses, both humoral and cellular, using a state of the art experimental approach. As such, the results are generally well supported by data and contribute significantly to filling a gap in our knowledge regarding the immune response of indigenous populations.

There are some issues that the authors may improve:

1. A key observation is that Australian First Nations people produce robust neutralizing antibodies to the Ancestral and Delta variants following vaccination, but with slightly reduced IgG titers. Figure 2 is devoted to understanding the mechanism that may explain the reduced IgG production by this indigenous population. The production of IgG is slightly higher in females (in both populations, although not reaching significance, Fig 2a). Can the fact that the group of Australian First Nations people contains only 47% of females, while the non-indigenous group contains 74% of females, contributes to a slightly higher production of IgG in the non-indigenous group (with more females)?
2. A very original finding is the relationship between glycosylation and reduced IgG response – described in Fig 2. The authors conclude that Australian First Nations people have a pattern of glycosylation, present before the first dose of the vaccine, that correlates with poor IgG production. However, my most significant concern related to this publication is that this pre-existent glycosylation pattern is associated with comorbidities and is not an intrinsic characteristic of Australian First Nations people (see Fig 2g). In fact, comparing panels 2d, 2g, and 2m, it appears that the distribution of Total G0 Abundance (or G0f) has a bimodal pattern among the Australian First Nations people (Fig 2d, with a proportion of the individuals with an abundance that overlaps with the non-indigenous group and another group of individuals with increased abundance); Fig 2g shows that almost all individuals with increased G0 abundance have comorbidities, which is further confirmed with Fig 2m where nearly all individuals with high G0 abundance have comorbidities (triangles, left panel), and the individuals with lower IgG titres (Fig 2m right panel) also have comorbidities (triangles). Therefore, it is hard to conclude that the observed reduced ability of Australian First Nations people to produce IgG is due to changes in the glycosylation pattern. The glycosylation pattern appears to be secondary to comorbidities, and the greater number of comorbidities within the Australian First Nations people leads to reduced IgG.
3. As a consequence of the point above, the correlation matrix in Fig 5 is very informative but it does not account for the types of associations described in the point above. Is it possible to find a correlation when correcting for these variables that appear to be associated (comorbidities, glycosylation, IgG production)?
4. The claim that IL-18 is overproduced in Australian First Nations people with comorbidities and correlated with G0 abundance (line 263), based on a comparison of the medians of IL-18, can be misleading. Figure 2l shows that there are only 4 individuals with comorbidities with very high levels of IL-18, while all others do not seem to have an IL-18 production significantly different from the other groups – it may be a consequence of specific comorbidities rather than a feature of Australian First Nations people. Furthermore, Fig 2m presents a correlation concerning comorbidities and non-comorbidities that can also mislead. Looking at the values with more extreme IL-18 production, they do not appear to be especially different from other cases of comorbidities (triangles) with lower IL-18 production regarding G0 abundance (left) or IgG production (right).
5. Given the importance of comorbidities in explaining an essential part of the divergent immune response of Australian First Nations people, the manuscript would benefit from data from non-indigenous people with comorbidities (like the data presented in Fig 2g) for other comparisons.
6. The consistency of reduced Tfh cells and reduced IgG production among Australian First Nations people is of great significance (Fig 3). A minor issue is a claim that the reduction of Tfh1 cells may

explain the low IgG production "Tfh type 1 cells facilitate key interactions with B cells" (line 583). Studies from Hideki Ueno showed that, unlike Tfh2 cells, the "Tfh1-like cells" could not provide help to B cells and appear to have a suppressive role.

7. Can the authors comment on the apparent contradiction between changes observed with tetramers (Fig 4) and the absence of consistent changes with stimulation-based assays (Fig 3)?

8. Among the study's limitations, the authors may consider mentioning that the study of Australian First Nations people following infection is biased towards severe disease.

Reviewer #3:

Remarks to the Author:

The study by Zhang et. al. and colleagues provides a comprehensive characterization of humoral and cellular immune profiles following SARS-CoV-2 vaccination and infection in Australian First Nations people. The authors find robust IgG neutralization titers after 2 doses of vaccine in First Nations participants, though it is significantly lower than non-indigenous people at peak response time point. Titers against Omicron variant and neutralizing antibody kinetics in indigenous population reported here is in line with previous studies; waning of titers after 6 months post vaccination followed by an increase after a 3rd booster dose. IgG titers negatively correlated with age which was driven mainly by participants with a reported comorbidity. The authors hypothesized that IgG Fc glycosylation differences at baseline might be a driver of differential antibody response and observed that First nations participants had significantly higher abundance of agalactosylated (G0) total IgG at baseline. The G0 abundance correlated negatively with IgG titers and positively with age. The higher G0 in the First nations people were driven primarily by people with comorbidities and was a feature of participants with comorbidities studied here irrespective of their ethnicity. In addition to higher IgG G0 at baseline, First nation people with comorbidities had higher levels of proinflammatory cytokines, especially IL-18 and lower frequencies of a) CD134+CD137+ Tfh responses at 28 days post second dose and b) spike specific IgD- B cells 6 months post vaccination. The authors also performed deep profiling of immune response in Australian First Nations people hospitalized COVID-19 patients. Overall, this study provides an extremely detailed characterization of immune responses in First nations people, however it does not clearly provide new biological insights.

Major points:

1. The authors have profiled total IgG fucosylation and galactosylation but have not shown any data on IgG sialylation. Since IgG sialylation is known to be strongly anti-inflammatory and can affect the quality of IgG response, an analysis on the same should be included in the data. Given that galactosylation is a prerequisite for IgG sialylation, higher G0 might translate to IgG sialylation abundance. This is an important piece of information, that is missing from the current data.

2. While the authors state that "Australian First Nations people with comorbidities showed reduced SARS-CoV-2 antibody axis linked to altered IgG glycosylation levels following BNT162b2 COVID-19 vaccination.", altered IgG G0 levels were found in subjects with comorbidities irrespective of ethnicities. Thus, validating the link between IgG G0 levels and IgG titers/response in another independent cohort will make their conclusion stronger.

Minor points:

1. In figure 1a, the formatting of the n for various time points are not properly aligned and also does not correlate with the data provided in the text.
2. Splitting up the age and gender in two panels, rather than combining them as done in figure 1c will be helpful for the readers.
3. Including data for participants with comorbidities in non-indigenous group in figure 2c will be informative.
4. The number of First nation participants receiving the ChAdOx1-S vaccine is low and does not add to the data presented here.

Author Rebuttal to Initial comments

RESPONSES TO REVIEWERS' COMMENTS

We immensely thank the Reviewers for their comments and insightful suggestions, which allowed us to greatly improve the manuscript. We also thank the Editors for the opportunity to submit the revised version of our manuscript to *Nature Immunology*.

Following the comments from the Editor and the Reviewers, we have substantially revised our manuscript, extended our COVID-19 vaccination cohort to non-Indigenous individuals with comorbidities and performed analyses on additional samples as suggested.

In short, we secured samples from additional 53 non-Indigenous individuals with co-morbidities: diabetes and renal disease (n=22) and inflammatory bowel disease (IBD; n=31) following COVID-19 vaccination (details listed in Supplementary Table 2). We performed a set of comprehensive analyses to define humoral, B cell and T cell responses in non-Indigenous participants with comorbidities following COVID-19 vaccination. Our additional analyses included anti-RBD antibody responses, glycosylation patterns of IgG antibodies, sialylation patterns for both First Nations and non-indigenous cohorts (as recommended by Reviewer 3), spike probe-specific B cell responses *ex vivo*, spike-specific T cells detected by the AIM assay and inflammation profiles for additional non-Indigenous participants with co-morbidities.

Our additional data provide evidence that non-Indigenous people with co-morbidities such as diabetes, renal function and IBD have decreased antibody responses associated with high abundance of G0 and increased IL-18 levels, in a similar pattern as First Nations people with comorbidities. Thus, these observed immune perturbations following COVID-19 vaccination are related to comorbidities rather than ethnicity, as we hypothesized in the first version of the manuscript.

Our data show a novel inverse correlation between SARS-CoV-2 antibody responses following COVID-19 vaccination and abundance of G0 within IgG antibodies at baseline for both First

Nations people as well as non-Indigenous people. Our findings can therefore be generalized across populations, however, are especially highly relevant to First Nations people who have disproportionate rates of comorbidities, including diabetes and renal disease. First Nations people across the world, such as Native American and Alaskan natives, also have disproportionate rates of diabetes, chronic respiratory and renal disease, with death rates from these chronic conditions 3-5 times higher than the non-Indigenous population, prior to COVID-19, as do Australian First Nations people.

Following the comments from Reviewer 1, we have also performed additional TCR $\alpha\beta$ sequencing and analyses to understand the uniqueness of TCR $\alpha\beta$ clonotypes within First Nations Australians.

Following the comments from Reviewer 2, we have performed in-depth multiple linear regression analyses to define correlations of COVID-19 vaccine immune responses.

Reviewer #1:

Remarks to the Author:

This is an interesting report that assesses the immune response to vaccination in a cohort of First Nation donors in Australia.

There is also assessment of the immune response to COVID-19 in hospital in a smaller group.

The work is important at this group has been at higher risk of COVID-19 over the last 2 years.

The work seeks to uncover the determinants that might underlie this risk.

The experimental work is very well done and the manuscript is very well written.

It is one of the most comprehensive studies of covid immunity in a cohort that I have seen.

The work is original and novel.

The major finding is that First Nation (FN) have higher levels of co-morbidity. These correlate with higher levels of non-glycosylated antibody and impaired humoral immunity.

We really thank the Reviewer for recognizing the importance and novelty of our research.

As 'co-morbidity' is such a key point in the article it would be important to know how this was assessed. It is stated as 'reported' by the subjects. How was this done, to what degree of granularity and was it objectively assessed. Could there be selective over-reporting of this phenotype in the FN group?

All participants were asked to verbally report comorbidities and for consent to review their medical records. Electronic medical records reviews were undertaken for each individual participant and evidence of the following co-morbidities searched for and documented: previous COVID-19, chronic respiratory disease, renal disease, diabetes, pregnancy, liver disease, returned traveller, cardiac disease, obesity BMI>30, neurological disease, nursing home resident, homeless, immunosuppressed or other (specified in free text).

As the co-morbidities information was objectively verified based on medical records for all participants irrespective of First Nations status, we do not think there is selective over-reporting of co-morbidities in the First Nations Group.

Following the Reviewer's comment, we included the above information in Methods (page 23):

“All COVAC participants were asked to verbally report comorbidities and for consent to review their medical records. Electronic medical records reviews were undertaken for each individual participant and evidence of the following comorbidities and factors were documented, including previous COVID-19, chronic respiratory disease, renal disease, diabetes, pregnancy, liver disease, returned traveller, cardiac disease, obesity BMI>30, neurological disease, nursing home resident, if homeless, immunosuppressed or other conditions/statuses (specified in free text).”

Perhaps the key point is whether or not the association of co-morbidity with altered 'antibody axis' is seen only in FN subjects or is true for all populations. As I understand it, there were not enough non-FN people with co-morbidity to definitively assess this. This needs further discussion to address.

We agree with the Reviewer that it is of great importance to understand whether the association with altered 'antibody axis' is also observed in non-Indigenous individuals with co-morbidities. As mentioned above, following the Reviewer's comment (as well as the comments from Reviewer 2 and 3), we have sourced additional samples from non-Indigenous individuals with co-morbidities following COVID-19 vaccination from 6 different cohorts and performed the additional experiments. In the revised version of our manuscript, we have analysed samples from additional (n=53) non-Indigenous individuals with co-morbidities, including diabetes and renal disease (n=22) and inflammatory bowel syndrome (n=31) following COVID-19 vaccination, in addition to the 16 samples from non-Indigenous individuals with co-morbidities for bulk IgG glycosylation analyses in the original submission.

These additional samples include all possible PBMC samples from individuals with co-morbidities such as diabetes and renal disease following COVID-19 vaccination available to us via our collaborative links. As the majority of the population have already been vaccinated (in Australia >95%), it was not feasible to recruit more vaccine-naïve patients with diabetes and renal disease to assess their responses to 1, 2 and 3 vaccine doses.

We have performed analyses to define humoral, B cell and T cell responses in these non-Indigenous participants with co-morbidities following COVID-19 vaccination. Our additional analyses included anti-RBD antibody responses, glycosylation patterns of IgG antibodies, sialylation patterns for both First Nations and non-Indigenous cohorts (as recommended by Reviewer 3), spike probe-specific B cell responses *ex vivo*, spike-specific T cells detected by the

AIM assay and inflammation profiles for additional non-Indigenous participants with comorbidities. Our data provide evidence that non-Indigenous people with co-morbidities such as diabetes, renal disease and IBD have decreased antibody responses associated with high abundance of G0, in a similar pattern as First Nations people.

We have modified our manuscript to include these additional data in Figure 3, described in Results (page 9-11):

“Reduced RBD-specific IgG titres and spike-specific B cell responses are associated with increased global IgG G0 levels in non-Indigenous individuals with comorbidities

Following our finding of such strong association between reduced SARS-CoV-2 antibody responses following COVID-19 vaccination and increased G0 abundance at baseline in First Nations people with comorbidities, we sought to define whether such correlation was related to ethnicity or, in contrast, linked predominantly to comorbidities. We obtained samples from an additional 69 non-Indigenous individuals with comorbidities, such as diabetes and renal disease (n=38) and inflammatory bowel disease (IBD; n=31), following COVID-19 vaccination (details listed in Supplementary Table 2). We performed analyses to define SARS-CoV-2 antibody and B cell responses in non-Indigenous participants with comorbidities following COVID-19 vaccination. Our additional analyses included anti-RBD antibody responses, glycosylation and sialylation patterns, spike probe-specific B cell responses *ex vivo* and inflammation profiles for additional non-Indigenous participants with comorbidities. Our data provide evidence that non-Indigenous people with comorbidities such as diabetes, renal disease and IBD also had decreased anti-RBD antibody responses (**Fig. 3a**), in a similar pattern as First Nations people with comorbidities. Similarly, compared to First Nations and non-Indigenous people without comorbidities, individuals with comorbidities had significantly elevated G0 abundance irrespective of ethnicity (**Fig. 3b**), although no association was observed with V3 RBD IgG titre (**Fig. 3c**). Similar to our observations in First Nations people with comorbidities, IL-18 levels at baseline were significantly increased in non-Indigenous individuals with comorbidities compared to those without (**Fig. 3d**), and inversely correlated with anti-RBD IgG titres in IBD patients (**Fig. 3e**). Our analyses also demonstrated reduced spike-specific B cell frequencies in non-Indigenous participants with IBD (**Fig. 3f**), thus supporting our antibody data as well as our findings in First Nations people with comorbidities.

We used multiple linear regression to investigate the contributions of First Nations status, comorbidities and G0 abundance to the (log) V3 anti-RBD IgG levels across the COVAC cohort as well as non-Indigenous cohorts with comorbidities. We found that comorbidities (p=0.014) and G0 abundance (p=0.041) were significant predictors of V3 responses to vaccination, whereas Indigenous status was not (p=1.0). Comorbidities remained a significant predictor after adjusting for age, gender and BMI (p=0.0067), but G0 abundance was not (p=0.17). This highlights the importance of comorbidities in determining the serological response to vaccination in both indigenous and non-indigenous populations.

Subsequently, to investigate the contribution of both demographic and immunological factors to (log) V3 RBD IgG responses to vaccination, we analysed the role of study, First Nations status, comorbidities, age, BMI, gender, V1 G0 proportion, and (log) V1 IL-18 levels. IL-18 was the most significant predictor of V3 anti-RBD IgG titres in this analysis ($p < 0.0001$), while only age ($p = 0.016$) and gender ($p = 0.035$) also remained significant. This demonstrates that IL-18 levels are an important predictor of vaccine responses.

Overall, our data show a novel inverse correlation between SARS-CoV-2 antibody responses following COVID-19 vaccination and G0 abundance in IgG antibodies at baseline for both First Nations people as well as non-Indigenous people. We provide evidence that reduced antibody axis linked with high G0 abundance at baseline is related to comorbidities rather than ethnicity, thus making G0 abundance (potentially together with IL-18) at baseline a biomarker for SARS-CoV-2-specific antibody and B cell responses following COVID-19 vaccination. Our findings are especially highly relevant to First Nations people as the prevalence of co-morbidities, including diabetes and renal disease is substantially higher within Indigenous populations globally. Similar to Australian First Nations people, Native American and Alaskan natives also have disproportionate rates of diabetes, chronic respiratory and renal disease, with death rates from these chronic conditions 3-5 times higher than non-Indigenous populations, prior to COVID-19⁹.

Figure 3. Reduced RBD-specific IgG titres and spike-specific B cells while increased global IgG G0 abundance in non-indigenous people with comorbidities. (a) End-point IgG titres of Ancestral RBD antibodies post-dose 2 COVID vaccine. (b) Abundance of total G0 and sialylation

G2S1f. (c) Correlation between total G0 abundance and Ancestral RBD IgG titres post-dose 2 COVID vaccine. (d) Concentration (pg/mL) of IL-18. (e) Correlation between IL-18 concentration and Ancestral RBD IgG titres post-dose 2 COVID vaccine. First Nations: $n_{\text{No comorbidities}}=28$, $n_{\text{With comorbidities}}=18$; non-Indigenous: $n_{\text{No comorbidities}}=39$, $n_{\text{With comorbidities}}=15$, $n_{\text{DISI}}=16$, $n_{\text{IBD}}=26$. Black horizontal dotted line indicates seropositivity defined as mean+2SD of pooled First Nations and non-Indigenous V1 IgG titre. (f) V4 frequency of spike-specific cells of IgD⁻ B cells. First Nations: $n_{\text{No comorbidities}}=7$, $n_{\text{With comorbidities}}=4$; non-Indigenous: $n_{\text{No comorbidities}}=13$, $n_{\text{With comorbidities}}=8$, $n_{\text{IBD}}=23$. Bars indicate median and boxes and whiskers indicate interquartile range, minimum and maximum. Statistical significance was determined with Mann-Whitney test. Exact P values are shown except **** $P<0.0001$. Correlation was determined with Spearman's correlation. Samples from non-Indigenous individuals with comorbidities were detailed in Methods.

Results (page 12):

Spike-specific CD4⁺ and CD8⁺ T cell responses detected by AIM assay (**Fig. 4g**) were comparable in both Australian First Nations participants and non-Indigenous individuals irrespective of comorbidities.

Figure 4. Prototypical CD4⁺ and CD8⁺ T cell responses in First Nations and non-Indigenous cohorts. (g-h) Frequency of spike-specific (g) activated and (h) IFN- γ ⁺TNF⁺ CD4⁺ and CD8⁺ T cells in individuals with or without comorbidities (First Nations: $n_{\text{No comorbidities}}=26$, $n_{\text{With comorbidities}}=15$; non-Indigenous: $n_{\text{No comorbidities}}=34$).

Discussion (page 19-20):

“Our data also demonstrated similar and significant alterations in basal IgG glycosylation patterns and antibody axis in non-Indigenous people with comorbidities, indicating that our findings relate to comorbidities across different ethnicities rather than First Nations ethnicity per se. As higher IgG G0 glycosylation levels correlated with lower RBD-specific IgG titres for both Australian

First Nations and non-Indigenous participants, markedly elevated bulk IgG G0 levels in Australian First Nations people can explain, at least in part, significantly reduced SARS-CoV-2-specific IgG levels after COVID-19 vaccination in Australian First Nations and non-Indigenous people with comorbidities. Altered IgG glycosylation levels in Australian First Nations and non-Indigenous people with comorbidities were also associated with increased basal levels of IL-18, which is one of the key cytokines related to severe COVID-19³². As agalactosylated IgG antibodies are regarded as pro-inflammatory³⁵ and were previously linked to lower antibody levels with reduced functionality in autoimmune cohorts following influenza vaccination³⁸, our study suggests that differential bulk IgG G0 antibody glycosylation profiles at baseline in Australian First Nations and non-Indigenous people with comorbidities can potentially underlie reduced SARS-CoV-2-specific IgG antibody levels and spike-specific B cell number following BNT162b2 mRNA vaccination. Our study also suggests that IgG G0 abundance prior to vaccination might potentially serve as a biomarker for humoral and B cell responses following COVID-19 vaccination, and possibly vaccinations against other infectious diseases.”

Methods (page 23):

“Details of COVID-19 vaccinated non-Indigenous participants with co-morbidities (diabetes and renal disease; n=22 individuals) as well as inflammatory bowel syndrome (n=31)⁷¹ are listed in Supplementary Table 2.”

The work on T cell responses is thorough. It was not entirely clear to me if the responses in the FN group could be because that population has a different profile of HLA alleles. I imagine these were not sequenced? This is discussed in the paper but could be made clearer.

We thank the Reviewer for this comment. We agree that differential HLA expression profiles in First Nations people could underlie differences in SARS-CoV-2 tetramer-specific CD8⁺ T cell responses, especially as all SARS-CoV-2 T cell epitopes, known to date, were identified previously in non-Indigenous individuals.

As T cell responses depend on the genetically determined, often ethnicity-specific, individual HLA profiles, immunogenic T cell peptides specific for HLAs prominent in non-Indigenous people might potentially be far from being immunodominant in First Nations individuals. As shown by our previous data (Clemens EB *et al*, Immunol & Cell Biol, 2016, PMID:26493179), Aboriginal and Torres Strait Islander have distinct HLA profiles, with 4 prevalent HLA-A types (HLA-A*24:02, HLA-A*11:01, HLA-A*34:01, HLA-A*02:01) and 4 prevalent HLA-B types (HLA-B*13:01, HLA-B*15:21; HLA-B*40:01/02, HLA-B*56:01/02). Based on our HLA analysis, the 5 most prominent HLAs in First Nations Australians (HLA-A*24:02, A*34:01, B*15:21, B*13:01, A*11:01) give a population coverage of ~82% for at least 1 HLA and ~63% for 2 HLA alleles. As, to date, SARS-CoV-2 epitopes have not been identified for HLAs highly specific for First Nations people (e.g HLA-B*13:01; HLA-A*34:01), it is thus possible that SARS-CoV-2 epitopes

restricted by the highly prominent HLAs in First Nations people could be immunodominant over CD8⁺ T cell specificities identified in non-Indigenous populations and used in our study. Thus, understanding immunogenicity and magnitude hierarchy of the SARS-CoV-2-derived peptides restricted by HLAs prominent in First Nations people and identified in First Nations individuals is of great importance for future studies.

We have sequenced HLAs for our COVAC cohort (data are provided in Supplementary Table 3).

Following the Reviewer's comments, we have modified the Results and Discussion.

Results (page 13-14):

“This could potentially be linked to the fact that the prominent A1/S₈₆₅, A2/S₂₆₉, A3/S₃₇₈ or A24/S₁₂₀₈ epitopes were previously identified in non-Indigenous individuals and hence might not be as immunodominant in Australian First Nations people expressing a different set of dominant HLA-I glycoproteins⁴⁹. As T cell responses depend on genetically determined (and often ethnicity-specific) individual HLA profiles, immunogenic T cell peptides specific for HLAs prominent in non-Indigenous people could potentially be far from being immunodominant in First Nations individuals. Thus, differential HLA expression profiles in First Nations people (Supplementary Table 3) could underlie differences in SARS-CoV-2 tetramer-specific CD8⁺ T cell responses, especially as all SARS-CoV-2 T cell epitopes, known to date, were previously identified in non-Indigenous individuals.

Discussion (page 18):

“Our data revealed prominent and comparable SARS-CoV-2-specific CD4⁺ and CD8⁺ T cell responses in Australian First Nations and non-Indigenous people following BNT162b2 vaccination, as detected by AIM and IFN- γ /TNF assays, irrespective of comorbidities. This finding indicates that even First Nations and non-Indigenous people with comorbidities have robust T cell responses following COVID-19 vaccination to provide at least some level of protection following subsequent SARS-CoV-2 infection. Similarly, detection of SARS-CoV-2-specific CD4⁺ T cells directly *ex vivo* with peptide-HLA tetramers showed robust CD4⁺ T cell responses directed against the prominent DPB4/S₁₆₇ epitope²³, suggesting immunodominance of DPB4/S₁₆₇⁺CD4⁺ T cells in both Australian First Nations people and non-Indigenous people.

In contrast to the comparable total SARS-CoV-2-specific CD8⁺ T cell responses detected in AIM and ICS assays (by culturing PBMCs with pooled spike peptides), our analysis of tetramer-specific CD8⁺ T cells directed at four SARS-CoV-2-specific CD8⁺ T cell epitopes (A1/S₈₆₅, A2/S₂₆₉, A3/S₃₇₈ and A24/S₁₂₀₈), previously identified as immunodominant in non-Indigenous people^{42, 43, 44, 46}, showed reduced frequencies of tetramer-specific CD8⁺ T cells in Australian First Nations people. Such discrepancy between CD8⁺ T cell data obtained from AIM/ICS assay and tetramer staining can potentially be explained by the fact that AIM/ICS assays detect total SARS-CoV-2-

specific T cell responses, whereas peptide-HLA tetramer staining is focused on epitope-specific T cells and is dependent on HLA distribution within individuals and/or ethnicities. Thus, differential HLA expression profiles in First Nations people could underlie differences in SARS-CoV-2 tetramer-specific CD8⁺ T cell responses, especially as all SARS-CoV-2 T cell epitopes, known to date, were identified previously in non-Indigenous individuals. As T cell responses depend on the genetically determined, often ethnicity-specific, individual HLA profiles, immunogenic T cell peptides specific for HLAs prominent in non-Indigenous people might potentially be far from being immunodominant in First Nations individuals. As shown by our previous data⁴⁹, Aboriginal and Torres Strait Islander have distinct HLA profiles, with 4 prevalent HLA-A types (HLA-A*24:02, HLA-A*11:01, HLA-A*34:01, HLA-A*02:01) and 4 prevalent HLA-B types (HLA-B*13:01, HLA-B*15:21; HLA-B*40:01/02, HLA-B*56:01/02). Based on our HLA analysis, the 5 most prominent HLAs in First Nations Australians (HLA-A*24:02, -A*34:01, -B*15:21, -B*13:01, -A*11:01) give a population coverage of ~82% for at least 1 HLA and ~63% for 2 HLA alleles. As, to date, SARS-CoV-2 epitopes have not been identified for HLAs highly specific for First Nations people (e.g HLA-B*13:01; HLA-A*34:01), it is therefore possible that SARS-CoV-2 epitopes restricted by the highly prominent HLAs in First Nations people could be immunodominant over CD8⁺ T cell specificities identified in non-Indigenous populations and used in our study. Thus, understanding immunogenicity and magnitude hierarchy of the SARS-CoV-2-derived peptides restricted by HLAs prominent in First Nations people and identified in First Nations individuals is a great importance for future studies, as exemplified by our previous work for influenza A and influenza B viruses^{64, 65}.”

The work on the hospitalised patients is very thorough but, sadly, I don't really know how it informs the reader. It could be considered for removal.

Following the Reviewer's comment, we have removed our work on First Nations people hospitalised with COVID-19.

The team is very comprehensive and talented. One of their talents is in TCR sequence analysis of epitope-specific responses. The emerging data to suggest that there is a potentially different profile of TCR using in the FN subjects is novel. If statistically confirmed this could be an important determinant of the functional quality of the T cell response. What is the statistical assessment of these findings?

We thank the Reviewer for their positive comments and agree that a potentially different TCR profile within HLA-A*02:02/S₂₆₉⁺CD8⁺ T cells is of great importance and could provide novel insights into TCR recognition of SARS-CoV-2 epitopes in First Nations people. Following the Reviewer's comment, we have performed additional experiments to single-cell sort additional HLA-A*02:02/S₂₆₉⁺CD8⁺ T cells for their TCRαβ sequencing to expand TCR numbers from First Nations people for our in-depth statistical analyses, which were performed by Drs Mikhail

Pogorelyy, Anastasia Minervina, Jeremy Chase Crawford and Paul Thomas at St Jude Children's Research Hospital, Memphis, TN, USA.

We have used our database of SARS-CoV-2-specific A2/S₂₆₉⁺CD8⁺ TCR clonotypes generated previously from non-Indigenous adults^{42, 43} to test whether certain V-segments were enriched or depleted between First Nations and Non-Indigenous people. We plotted the frequencies of V segments as a scatterplot across these two cohorts (**Supplementary Fig. 4a**). Our analysis revealed that TRAV12-1 is far below the equivalence line and underrepresented in First Nations individuals, and there was a statistically significant difference in a non-random association between TRAV12-1 across the two groups ($p=0.01856$; Fisher's Exact test).

Supplementary Fig. 4. Statistical analyses of TCR gene segment usage in First Nations and non-Indigenous individuals. (a) Scatter plot of frequency of TRAV segment in A2/S₂₆₉-specific clonotypes by population.

Furthermore, in contrast to a strong TCR α motif detected for A2/S₂₆₉-specific TRAV12-1 in non-Indigenous adults, there was a lack of any statistically significant ($\chi^2 > 50$) TRAV motifs among First Nations A2/S₂₆₉-specific TCRs (**Supplementary Fig. 4b**, top). Despite the lack of a TRAV12 motif in the First Nations A2/S₂₆₉ response, we still identified the concordant TRBV2-2 motif in both non-indigenous and First Nations individuals, indicating that core features of the A2/S₂₆₉ response are conserved across these two populations.

Supplementary Fig. 4 (b) A subset of statistically significant motifs detected for the A2 response. NIA, Non-Indigenous Adults; FN, First Nations Adults. ‘A’ indicates alpha chain, ‘B’ indicates beta chain.

We subsequently asked what TCR α and TCR β segments were enriched in First Nations A2/S₂₆₉-specific TCRs. In contrast to a strong enrichment of TRAV12-1 significantly pairing with TRAJ43 in non-Indigenous adults (**Supplementary Fig. 4c**)^{44,45}, we found that among the First Nation A2/S₂₆₉-specific TCRs, although TRAV12-1 was present, it was substantially less enriched (**Supplementary Fig. 4c**). For non-Indigenous adults, TRAV12-1 had a log₂ fold change >8 compared to a random human repertoire, whereas for First Nation individuals, TRAV12-1 had only a log₂ fold change of ~2 which was greater than background. While there is a statistically significant enriched junction between TRAV12-1/TRAJ43 among non-Indigenous adult A2/S₂₆₉-specific TCRs, no significant junction exists among First Nations A2/S₂₆₉-specific TCRs. Importantly, TRAV12-1 can be detected within A2/S₂₆₉-specific TCRs in both populations and the typical TRBJ2-2 enrichment can also be observed across both groups, which help us to exclude a technical nor analytical artifact.

Supplementary Fig. 4 (c) Cord plots depicting v-segment and j-segment pairings across TCR clonotypes for the A2 response for NIA (top) and FN (bottom) populations.

In accordance with the findings above, the First Nations A2/S₂₆₉-specific alpha chains did not resemble the epitope-specific TCR repertoire analyzed by our typical TCRdist approach, although other First Nations SARS-CoV-2-specific epitopes do, for example DP4/S₁₆₇ (**Supplementary Fig. 4d**, left). However, the A2/S₂₆₉-specific beta chains did exhibit patterns indicative of epitope specificity in both populations (**Supplementary Fig. 4d**, middle), in part driven by the TRBJ2-2 motif identified above.

Supplementary Fig. 4 (d) Smoothed frequency distributions of averaged nearest-neighbors TCRdist scores across NIA and FN populations for distinct epitopes. Left: alpha chains; middle: beta chains; right: paired alpha-beta chains. Numerical values in the legend correspond to the average neighbor distance for that repertoire. Lower values on the x-axis represent more similarity across TCR clonotypes.

We have included the above analyses in Results (page 15-16):

“Using our database of SARS-CoV-2-specific A2/S₂₆₉⁺CD8⁺ TCR clonotypes generated previously from non-Indigenous adults^{42, 43}, we tested whether certain V-segments were enriched or depleted between First Nations and non-Indigenous people within A2/S₂₆₉⁺CD8⁺ TCRs. A Fisher's exact test was used to determine that the TRAV12-1*01 frequency within the First Nations response was statistically distinct from that of non-Indigenous adults ($P=0.019$). We observed a statistically enriched TCR α -chain motif in the non-Indigenous A2/S₂₆₉ response, largely driven by TRAV12-1 usage that was not observed in the First Nations A2/S₂₆₉ TCR repertoire (**Supplementary Fig. 4b**, top). However, we also observed similar TCR β -chain motifs in the A2/S₂₆₉ repertoire between First Nations and non-Indigenous participants largely driven by TRBJ2-2, suggesting that core features of the A2/S₂₆₉-specific TCR repertoire remain preserved between these two populations (**Supplementary Fig. 4b** middle, bottom). Again, the primary difference in the A2/S₂₆₉-specific TCR repertoire between these two populations seems to largely hinge on the extreme enrichment of the TRAV12-1 segment within the non-Indigenous population (>8 log fold change over expectations in a random TCR repertoire) compared to the First Nations people (~2 log fold change over the null expectation), despite the typical pairing of TRAV12-containing α -chains with TRBJ2-containing β -chains in the non-indigenous response (**Supplementary Fig. 4c**). This difference resulted in an TCR α -chain A2/S₂₆₉ repertoire that was structurally very diverse among the First Nations people compared to that of the non-Indigenous population (**Supplementary Fig. 4d**, left, comparing dark blue to light blue distributions). However, the TCR β -chain A2/S₂₆₉-specific repertoire exhibited comparable levels of epitope-

specific TCR similarity across these two populations (**Supplementary Fig. 4d**, middle), in part due to the TRBJ2-2 motif identified in both populations.”

The authors may wish to allude as to the degree to which this effect is genetic or acquired. Is there allelic variation in germline TCR V or J regions in FN? I assume not, and that this represents a selective difference during clonal evolution.

We also agree with the Reviewer that understanding allelic variation in germline TCR V or J regions in First Nations people is of great interest. Following the Reviewer’s comment, we analysed bulk RNAseq sample datasets from three First Nations Australians previously published <https://www.ahajournals.org/doi/full/10.1161/CIRCULATIONAHA.118.033891>. Specifically, we analyzed samples with SRA accession numbers SRR2748664, SRR2748668 and SRR2748690 (unstimulated PBMCs RNAseq). We used the bowtie2 program to build the reference from IMGT TRAV12-1*01, TRAV12-2*01 and TRAV12-3*01 alleles and then aligned all the reads to that reference. The resulting alignment covered nearly the entire V segment (with less coverage at the 5’ end), and it showed that in all First Nations people there were no mismatches to the TRAV12-1*01 allele from IMGT. In the SRR2748690 donor, there was an allelic variant in TRAV12-2, but this was a known TRAV12-2*02 variant.

Although there were no differences in coding regions, there was a possibility that differences in non-coding regions could significantly affect TRAV12-1 usage. To determine if this was the case, we reconstructed the TCR alpha repertoire from the same RNAseq data using mixcr to estimate V-usage of TRAV12-1. There were no marked differences in segment usage compared to what we previously observed in other datasets. 12 TCR alpha clones used TRAV12-1 out of 369 TCR alpha clones (0.033); 6 out of 137 (0.043); and 20 out of 349 (0.057), for SRR2748664, SRR2748668 and SRR2748690, respectively. In previously published TCR alpha repertoires from a cohort from Nicaragua (<https://elifesciences.org/articles/73475>), the median usage of TRAV12-1 was 0.034, which is very similar to what we observed here.

To summarize, our analysis of public RNAseq data showed that there were no mutations in the germline CDR1/CDR2 regions or other parts of TRAV12-1 sequence, and the usage of this segment in bulk repertoire is similar to other datasets.

We have referred to this analysis in Results (page 16):

“In addition, our analysis of public RNAseq dataset containing bulk RNAseq samples from three First Nations Australians⁵⁷ showed that there were no mutations in the germline CDR1/CDR2 regions or other parts of TRAV12-1 sequence, and that the usage of this germline segment in bulk TCR repertoire is the same as in other datasets⁵⁸.”

The article is very long and the figures are extensive. I am surprised that this length of article could be considered for print publication. There is a case for rationalising the text and figures.

We have removed our work on First Nations people hospitalised with COVID-19, as per the Reviewer's recommendation as well as rationalized the text. Following the comment from Reviewer 3, we have also removed ChAdOx1-S vaccine data from our manuscript.

Reviewer #2:

Remarks to the Author:

The manuscript by Zhang and colleagues describes the immune responses in Australian First Nations people towards vaccination with the COVID-19 BNT162b2 and following COVID-19 infection in non-exposed individuals. This report is of great significance since COVID-19 affected indigenous populations in very different geographies with particular severity. Yet, despite the disease burden, these populations' immune responses have remained mostly unaddressed. The study provides a comprehensive overview of immune responses, both humoral and cellular, using a state of the art experimental approach. As such, the results are generally well supported by data and contribute significantly to filling a gap in our knowledge regarding the immune response of indigenous populations.

We immensely thank the Reviewer for recognizing the importance and novelty of our research.

There are some issues that the authors may improve:

- 1. A key observation is that Australian First Nations people produce robust neutralizing antibodies to the Ancestral and Delta variants following vaccination, but with slightly reduced IgG titers. Figure 2 is devoted to understanding the mechanism that may explain the reduced IgG production by this indigenous population. The production of IgG is slightly higher in females (in both populations, although not reaching significance, Fig 2a). Can the fact that the group of Australian First Nations people contains only 47% of females, while the non-indigenous group contains 74% of females, contributes to a slightly higher production of IgG in the non-indigenous group (with more females)?*

We have plotted IgG titres according to gender (Fig. 2a) and performed a 2-way ANOVA analysis. Our analysis shows that while gender had an effect on antibody titre ($P=0.0400$), ethnicity status had a significant and even stronger effect ($P=0.0018$). No interaction effect was observed ($P=0.6835$). The factors leading to different antibody titre between ethnicity status were further investigated in the following sections.

Figure 2. Higher bulk IgG G0 abundance observed in First Nations individuals with renal disease and diabetes and associated with lower RBD IgG titre. (a) V3 Ancestral RBD IgG titre correlations with age ($n_{\text{First Nations}}=46$, $n_{\text{non-Indigenous}}=39$) and BMI ($n_{\text{First Nations}}=39$, $n_{\text{non-Indigenous}}=36$) and grouped by gender (First Nations: $n_{\text{Male}}=22$, $n_{\text{Female}}=24$; non-Indigenous: $n_{\text{Male}}=10$, $n_{\text{Female}}=29$).

This is now included in Results (page 7):

“While there were more females in non-Indigenous cohort (**Fig 1c**), 2-way ANOVA analysis revealed significant ($P=0.0018$) difference in antibody titres between First Nations and non-Indigenous cohorts when accounting gender (**Fig. 2a**).”

2. A very original finding is the relationship between glycosylation and reduced IgG response – described in Fig 2. The authors conclude that Australian First Nations people have a pattern of glycosylation, present before the first dose of the vaccine, that correlates with poor IgG production. However, my most significant concern related to this publication is that this pre-existent glycosylation pattern is associated with comorbidities and is not an intrinsic characteristic of Australian First Nations people (see Fig 2g). In fact, comparing panels 2d, 2g, and 2m, it appears that the distribution of Total G0 Abundance (or G0f) has a bimodal pattern among the Australian First Nations people (Fig 2d, with a proportion of the individuals with an abundance that overlaps with the non-indigenous group and another group of individuals with increased abundance); Fig 2g shows that almost all individuals with increased G0 abundance have comorbidities, which is further confirmed with Fig2m where nearly all individuals with high G0 abundance have comorbidities (triangles, left panel), and the individuals with lower IgG titres (Fig 2m right panel) also have comorbidities (triangles). Therefore, it is hard to conclude that the observed reduced ability of Australian First Nations people to produce IgG is due to changes in the glycosylation pattern. The glycosylation pattern appears to be secondary to comorbidities, and the greater number of comorbidities within the Australian First Nations people leads to reduced IgG.

We wholeheartedly agree with the Reviewer. As we outlined at the start of this Rebuttal Letter, we have secured samples from an additional 53 non-Indigenous individuals with co-morbidities: diabetes and renal disease (n=22) and inflammatory bowel disease (IBD; n=31) following COVID-19 vaccination (details listed in Supplementary Table 2). These additional samples include all possible PBMC samples from individuals with diabetes and renal disease following COVID-19 vaccination available to us via our collaborative links.

We have performed analyses to define humoral, B cell and T cell responses in these non-Indigenous participants with co-morbidities following COVID-19 vaccination. Our additional analyses included anti-RBD antibody responses, glycosylation patterns of IgG antibodies, sialylation patterns for both First Nations and non-Indigenous cohorts (as recommended by Reviewer 3), spike probe-specific B cell responses *ex vivo*, spike-specific T cells detected by the AIM assay and inflammation profiles for additional non-indigenous participants with co-morbidities. Our data provide evidence that non-Indigenous people with co-morbidities such as diabetes, renal function and IBD have decreased antibody responses associated with high abundance of G0, in a similar pattern as the First Nations people.

We would like to refer the Reviewer to our detailed description of our additional data, together with the figures, in our response to Reviewer 1 (comment #2, pages 2-6, starting with “Perhaps the key point is...”).

3. As a consequence of the point above, the correlation matrix in Fig 5 is very informative but it does not account for the types of associations described in the point above. Is it possible to find a correlation when correcting for these variables that appear to be associated (comorbidities, glycosylation, IgG production)?

Following the Reviewer’s comment, we have sought expertise from Drs Miles Davenport and David Khoury from the Kirby Institute (Sydney, NSW, Australia), who performed in-depth multiple linear regression analyses to define correlations of COVID-19 vaccine immune responses.

The Reviewer raises an important question regarding the relative roles of First Nations status and comorbidities in determining the difference in vaccine outcomes. However, we are unclear of the specific question raised and think that this is a request to examine the predictive value of [*First Nations status, comorbidities, glycosylation*] on [*vaccination outcome (V3 anti-RBD IgG)*]. We have addressed this in two ways: firstly, by including the non-Indigenous subjects with comorbidities (i) diabetes and renal disease; (ii) IBD; and secondly, by performing multiple linear regression on these factors.

Analysis of these predictors across the combined cohorts suggests that comorbidity is a significant predictor (p=0.0142), as is G0 abundance (p=0.041). Conversely, First Nations status is not a significant factor (p=1.0).

This is now included in Results (page 10):

“We used multiple linear regression to investigate the contributions of First Nations status, comorbidities and G0 abundance to the (log) V3 anti-RBD IgG levels across the COVAC cohort as well as non-Indigenous cohorts with comorbidities. We found that comorbidities ($P=0.014$) and G0 abundance ($P=0.041$) were significant predictors of V3 responses to vaccination, whereas Indigenous status was not ($P=1.0$). Comorbidities remained a significant predictor after adjusting for age, gender and BMI ($P=0.0067$), but G0 abundance was not ($P=0.17$). This highlights the importance of comorbidities in determining the serological response to vaccination in both Indigenous and non-Indigenous populations.”

Similarly, we expanded this list to include other demographic factors such as age, gender, and BMI as well as IL-18 levels. In this combined model, IL-18 levels emerged as the most significant predictor, with age and gender being the only other significant predictors. We have now included this in the results section (page 10):

“Subsequently, to investigate the contribution of both demographic and immunological factors to (log) V3 RBD IgG responses to vaccination, we analysed the role of First Nations status, comorbidities, age, BMI, gender, V1 G0 proportion, and (log) V1 IL-18 levels. IL-18 was the most significant predictor of V3 anti-RBD IgG titres in this analysis ($P<0.0001$), while only age ($P=0.016$) and gender ($P=0.035$) also remained significant. This demonstrates that IL-18 levels are an important predictor of vaccine responses.”

We thank the Reviewer for suggesting these analyses, which we believe have helped to clarify and define the contribution of different factors to COVID-19 vaccine immune responses.

4. The claim that IL-18 is overproduced in Australian First Nations people with comorbidities and correlated with G0 abundance (line 263), based on a comparison of the medians of IL-18, can be misleading. Figure 2l shows that there are only 4 individuals with comorbidities with very high levels of IL-18, while all others do not seem to have an IL-18 production significantly different from the other groups – it may be a consequence of specific comorbidities rather than a feature of Australian First Nations people. Furthermore, Fig 2m presents a correlation concerning comorbidities and non-comorbidities that can also mislead. Looking at the values with more extreme IL-18 production, they do not appear to be especially different from other cases of comorbidities (triangles) with lower IL-18 production regarding G0 abundance (left) or IgG production (right).

As stated above, we have performed the additional experiment to address whether the pre-existing high G0 abundance and high IL-18 levels were related to First Nations ethnicities or comorbidities. Our data provide clear evidence that non-Indigenous people with co-morbidities such as diabetes,

renal function and IBD have decreased antibody responses associated with high abundance of G0, in a similar pattern as the First Nations people. Our findings, however, remain highly relevant to First Nations people who have disproportionate rates of comorbidities, especially diabetes and renal disease.

5. Given the importance of comorbidities in explaining an essential part of the divergent immune response of Australian First Nations people, the manuscript would benefit from data from non-indigenous people with comorbidities (like the data presented in Fig 2g) for other comparisons.

In our modified version of the manuscript, we have included additional data on immune responses and glycosylation patterns in non-Indigenous people with co-morbidities following COVID-19 vaccination (please refer to our data and detailed description above).

6. The consistency of reduced Tfh cells and reduced IgG production among Australian First Nations people is of great significance (Fig 3). A minor issue is a claim that the reduction of Tfh1 cells may explain the low IgG production "Tfh type 1 cells facilitate key interactions with B cells" (line 583). Studies from Hideki Ueno showed that, unlike Tfh2 cells, the "Tfh1-like cells" could not provide help to B cells and appear to have a suppressive role.

We thank the Reviewer for this comment. We have modified the above-mentioned sentence to read:

“Reduced SARS-CoV-2 RBD antibodies towards the Ancestral and Delta strains were associated with significant decreases in spike-specific B cells (producing antibodies) and Tfh type 1 cells involved in interactions with B cells.”

7. Can the authors comment on the apparent contradiction between changes observed with tetramers (Fig 4) and the absence of consistent changes with stimulation-based assays (Fig 3)?

Following the comments from both Reviewer 1 and 2, we have modified the Results and Discussion to describe contradictions between T cell responses detected by AIM/ICS assays and tetramers.

Results (page 13):

“This could be potentially linked to the fact that the prominent A1/S₈₆₅, A2/S₂₆₉, A3/S₃₇₈ or A24/S₁₂₀₈ epitopes were previously identified in non-Indigenous individuals and hence might not be immunodominant in Australian First Nations people expressing a different set of dominant HLA-I glycoproteins⁴⁹. As T cell responses depend on the genetically determined (and often

ethnicity-specific) individual HLA profiles, immunogenic T cell peptides specific for HLAs prominent in non-Indigenous people could potentially be far from being immunodominant in First Nations individuals. Thus, differential HLA expression profiles in First Nations people (Supplementary Table 3) could underlie differences in SARS-CoV-2 tetramer-specific CD8⁺ T cell responses, especially as all SARS-CoV-2 T cell epitopes, known to date, were identified previously in non-Indigenous individuals.

Discussion (page 18):

“Our data revealed prominent and comparable SARS-CoV-2-specific CD4⁺ and CD8⁺ T cell responses in Australian First Nations and non-Indigenous people following BNT162b2 vaccination, as detected by AIM and IFN- γ /TNF assays, irrespective of co-morbidities. This finding indicates that First Nations and non-Indigenous people with co-morbidities have robust T cell responses following COVID-19 vaccination to provide at least some level of protection following subsequent SARS-CoV-2 infection. Similarly, detection of SARS-CoV-2-specific CD4⁺ T cells directly *ex vivo* with peptide-HLA tetramers showed robust CD4⁺ T cell responses directed against the prominent DPB4/S₁₆₇ epitope²³, suggesting immunodominance of DPB4/S₁₆₇⁺CD4⁺ T cells in both Australian First Nations people and non-Indigenous people.

In contrast to the comparable total SARS-CoV-2-specific CD8⁺ T cell responses detected in AIM and ICS assays (by culturing PBMCs with pooled spike peptides), our analysis of tetramer-specific CD8⁺ T cells directed at four SARS-CoV-2-specific CD8⁺ T cell epitopes (A1/S₈₆₅, A2/S₂₆₉, A3/S₃₇₈ and A24/S₁₂₀₈), previously identified as immunodominant in non-Indigenous people^{44,45,46,48}, showed reduced frequencies of tetramer-specific CD8⁺ T cells in Australian First Nations people. Such a discrepancy between CD8⁺ T cell data obtained from AIM/ICS assays and tetramer staining can potentially be explained by the fact that AIM/ICS assays detect total SARS-CoV-2-specific T cell responses, whereas peptide-HLA tetramer staining is focused on epitope-specific T cells and is dependent on HLA distribution within individuals and/or ethnicities. Thus, differential HLA expression profiles in First Nations people could underlie differences in SARS-CoV-2 tetramer-specific CD8⁺ T cell responses, especially as all SARS-CoV-2 T cell epitopes, known to date, were identified previously in non-Indigenous individuals. As T cell responses depend on the genetically determined, often ethnicity-specific, individual HLA profiles, immunogenic T cell peptides specific for HLAs prominent in non-Indigenous people might potentially be far from being immunodominant in First Nations individuals. As shown by our previous data⁴⁹, Aboriginal and Torres Strait Islander have distinct HLA profiles, with 4 prevalent HLA-A types (HLA-A*24:02, HLA-A*11:01, HLA-A*34:01, HLA-A*02:01) and 4 prevalent HLA-B types (HLA-B*13:01, HLA-B*15:21; HLA-B*40:01/02, HLA-B*56:01/02). Based on our HLA analysis, the 5 most prominent HLAs in First Nations Australians (HLA-A*24:02, -A*34:01, -B*15:21, -B*13:01, -A*11:01) give a population coverage of ~82% for at least 1 HLA and ~63% for 2 HLA alleles. As, to date, SARS-CoV-2 epitopes have not been identified for HLAs highly specific for First Nations people (e.g HLA-B*13:01; HLA-A*34:01), it is therefore possible

that SARS-CoV-2 epitopes restricted by the highly prominent HLAs in First Nations people could be immunodominant over CD8⁺ T cell specificities identified in non-Indigenous populations and used in our study. Thus, understanding immunogenicity and magnitude hierarchy of the SARS-CoV-2-derived peptides restricted by HLAs prominent in First Nations people and identified in First Nations individuals is of great importance for future studies, as exemplified by our previous work for influenza A and influenza B viruses^{64,65}.”

8. *Among the study's limitations, the authors may consider mentioning that the study of Australian First Nations people following infection is biased towards severe disease.*

We agree with the Reviewer. However, following the comment from Reviewer 1, we have removed the data related to immune responses in First Nations people following SARS-CoV-2 infection from the manuscript.

Reviewer #3:

Remarks to the Author:

The study by Zhang et. al. and colleagues provides a comprehensive characterization of humoral and cellular immune profiles following SARS-CoV-2 vaccination and infection in Australian First Nations people. The authors find robust IgG neutralization titers after 2 doses of vaccine in First Nations participants, though it is significantly lower than non-indigenous people at peak response time point. Titers against Omicron variant and neutralizing antibody kinetics in indigenous population reported here is in line with previous studies; waning of titers after 6 months post vaccination followed by an increase after a 3rd booster dose. IgG titers negatively correlated with age which was driven mainly by participants with a reported comorbidity. The authors hypothesized that IgG Fc glycosylation differences at baseline might be a driver of differential antibody response and observed that First nations participants had significantly higher abundance of agalactosylated (G0) total IgG at baseline. The G0 abundance correlated negatively with IgG titers and positively with age. The higher G0 in the First nations people were driven primarily by people with comorbidities and was a feature of participants with comorbidities studied here irrespective of their ethnicity. In addition to higher IgG G0 at baseline, First nation people with comorbidities had higher levels of proinflammatory cytokines, especially IL-18 and lower frequencies of a) CD134+CD137+ Tfh responses at 28 days post second dose and b) spike specific IgD- B cells 6 months post vaccination. The authors also performed deep profiling of immune response in Australian First Nations people hospitalized COVID-19 patients. Overall, this study provides an extremely detailed characterization of immune responses in First nations people, however it does not clearly provide new biological insights.

We greatly thank the Reviewer for recognizing our thorough characterisation of humoral and T cell responses following BNT162b2 vaccination in Australia First Nations people. However, we respectfully disagree that this work does not provide new biological insights since 1) we revealed G0 abundance and IL-18 level as important predictors of vaccine responses after regressing out demographic factors; 2) we observed significant differences in TRAV usage of A2/S₁₆₉⁺CD8⁺ T cells in Australia First Nations people.

Major points:

1. The authors have profiled total IgG fucosylation and galactosylation but have not shown any data on IgG sialylation. Since IgG sialylation is known to be strongly anti-inflammatory and can affect the quality of IgG response, an analysis on the same should be included in the data. Given that galactosylation is a prerequisite for IgG sialylation, higher G0 might translate to IgG sialylation abundance. This is an important piece of information, that is missing from the current data.

We agree with the Reviewer and following this, we have performed the additional experiments to assess IgG sialylation abundance. Non-surprisingly, our data show lower G2S1f level in First Nations individuals with comorbidities ($P < 0.0001$).

Figure 2. Higher bulk IgG G0 abundance observed in First Nations individuals with renal disease and diabetes and associated with lower RBD IgG titre. (g) V1 abundance of total G0, total fucose and sialylation G2S1f in individuals with or without comorbidities (First Nations: $n_{\text{No comorbidities}}=15$, $n_{\text{With comorbidities}}=11$; non-Indigenous: $n_{\text{No comorbidities}}=33$).

2. While the authors state that “Australian First Nations people with comorbidities showed reduced SARS-CoV-2 antibody axis linked to altered IgG glycosylation levels following BNT162b2

COVID-19 vaccination.”, altered IgG G0 levels were found in subjects with comorbidities irrespective of ethnicities. Thus, validating the link between IgG G0 levels and IgG titers/response in another independent cohort will make their conclusion stronger.

We are in total agreement with the Reviewer and have performed additional experiments in non-Indigenous cohorts with co-morbidities. As we outlined at the beginning of our Rebuttal Letter, we have secured samples from additional 53 non-Indigenous individuals with co-morbidities: diabetes and renal disease (n=22) and inflammatory bowel disease (IBD; n=31) following COVID-19 vaccination (details listed in Supplementary Table 2). These additional samples include all possible PBMC samples from individuals with co-morbidities following COVID-19 vaccination available to us via our collaborative links.

We have performed analyses to define humoral, B cell and T cell responses in these non-Indigenous participants with co-morbidities following COVID-19 vaccination. Our additional analyses included anti-RBD antibody responses, glycosylation patterns of IgG antibodies, sialylation patterns for both First Nations and non-Indigenous cohorts, spike probe-specific B cell responses *ex vivo*, spike-specific T cells detected by the AIM assay and inflammation profiles for additional non-Indigenous participants with co-morbidities. Our data provide evidence that non-Indigenous people with co-morbidities such as diabetes, renal function and IBD have decreased antibody responses associated with high abundance of G0, in a similar pattern as the First Nations people.

A more detailed description of our additional data, together with the figures, is also provided above in our response to Reviewer 1, comment #2, pages 2-6, starting with “Perhaps the key point is...”.

Minor points:

1. In figure 1a, the formatting of the n for various time points are not properly aligned and also does not correlate with the data provided in the text.

We have aligned the graph and updated the n numbers following removing ChAdOx1-S vaccine participants.

Figure 1. Robust IgG responses towards Ancestral and Delta RBD in First Nations and non-Indigenous cohorts following BNT162b2 vaccination. (a) COVAC cohort study design and sample collection.

2. Splitting up the age and gender in two panels, rather than combining them as done in figure 1c will be helpful for the readers.

We have split the age and gender into two panels.

Figure 1 (c) Age and gender (%) of COVAC participants receiving BNT162b2.

3. Including data for participants with comorbidities in non-indigenous group in figure 2c will be informative.

We have included data for participants with comorbidities in non-Indigenous group. Please refer to our responses above on pages 2-6.

4. The number of First nation participants receiving the ChAdOx1-S vaccine is low and does not add to the data presented here.

Following the Reviewer's comments, we have removed ChAdOx1-S vaccine data from the manuscript.

Decision Letter, first revision:

10th Feb 2023

Dear Dr. Kedzierska,

Thank you for submitting your revised manuscript "Robust and prototypical immune responses towards COVID-19 BNT162b2 vaccine in First Nations people are impacted by comorbidities" (NI-A34730A). It has now been seen by the original referees and their comments are below. We are happy to inform you that if you revise your manuscript appropriately according to our editorial requirements, your manuscript should be publishable in Nature Immunology.

I will now pre-edit the current version of your paper. We will also perform detailed checks on your paper and will send you a checklist detailing our editorial and formatting requirements in about two weeks. Please do not upload the final materials and make any revisions until you receive this additional information from us.

In the meantime however, please deposit all omics and code data into public repositories so that the accession codes are readily available to be added in the revised manuscript. We cannot accept the paper without them. In addition, please check that your ORCID is linked to your Nature account, as this frequently causes delays at acceptance. Should you have any query or comments about ORCID, please do not hesitate to contact our editorial assistant at immunology@us.nature.com.

If you had not uploaded a Word file for the current version of the manuscript, we will need one before beginning the editing process; please email that to immunology@us.nature.com at your earliest convenience.

Thank you again for your interest in Nature Immunology. Please do not hesitate to contact me if you have any questions.

Sincerely,

Ioana Visan, Ph.D.
Senior Editor
Nature Immunology

Tel: 212-726-9207
Fax: 212-696-9752
www.nature.com/ni

Reviewer #2 (Remarks to the Author):

The manuscript was significantly improved by including new data and a more thorough discussion. All issues raised in the previous report were addressed. It is a contribution of great significance in addressing in great detail the immune response of indigenous populations to SARS-CoV-2 infection, a population that has been affected disproportionately by severe COVID-19.

Reviewer #3 (Remarks to the Author):

The authors adequately addressed the comments and I support publication.

Final Decision Letter:

Dear Dr. Kedzierska,

I am delighted to accept your manuscript entitled "Robust and prototypical immune responses towards COVID-19 vaccine in First Nations peoples are impacted by comorbidities" for publication in an upcoming issue of Nature Immunology.

Over the next few weeks, your paper will be copyedited to ensure that it conforms to Nature Immunology style. Once your paper is typeset, you will receive an email with a link to choose the appropriate publishing options for your paper and our Author Services team will be in touch regarding any additional information that may be required.

Please note that *Nature Immunology* is a Transformative Journal (TJ). Authors may publish their research with us through the traditional subscription access route or make their paper immediately open access through payment of an article-processing charge (APC). Authors will not be required to make a final decision about access to their article until it has been accepted. [Find out more about Transformative Journals](https://www.springernature.com/gp/open-research/transformative-journals).

Authors may need to take specific actions to achieve  > **compliance** **with funder and institutional open access mandates**. If your research is supported by a funder that requires immediate open access (e.g. according to Plan S principles) then you should select the gold OA route, and we will direct you to the compliant route where possible. For authors selecting the subscription publication route, the journal's standard licensing terms will need to be accepted, including self-archiving policies. Those licensing terms will supersede any other terms that the author or any third party may assert apply to any version of the manuscript.

Your paper will be published online soon after we receive your corrections and will appear in print in the next available issue. Content is published online weekly on Mondays and Thursdays, and the embargo is set at 16:00 London time (GMT)/11:00 am US Eastern time (EST) on the day of publication. Now is the time to inform your Public Relations or Press Office about your paper, as they might be interested in promoting its publication. This will allow them time to prepare an accurate and satisfactory press release. Include your manuscript tracking number (NI-A34730B) and the name of the journal, which they will need when they contact our office.

About one week before your paper is published online, we shall be distributing a press release to news organizations worldwide, which may very well include details of your work. We are happy for your institution or funding agency to prepare its own press release, but it must mention the embargo date and Nature Immunology. Our Press Office will contact you closer to the time of publication, but if you or your Press Office have any enquiries in the meantime, please contact press@nature.com.

Also, if you have any spectacular or outstanding figures or graphics associated with your manuscript - though not necessarily included with your submission - we'd be delighted to consider them as candidates for our cover. Simply send an electronic version (accompanied by a hard copy) to us with a possible cover caption enclosed.

If you have not already done so, we strongly recommend that you upload the step-by-step protocols used in this manuscript to the Protocol Exchange. Protocol Exchange is an open online resource that allows researchers to share their detailed experimental know-how. All uploaded protocols are made freely available, assigned DOIs for ease of citation and fully searchable through nature.com. Protocols

can be linked to any publications in which they are used and will be linked to from your article. You can also establish a dedicated page to collect all your lab Protocols. By uploading your Protocols to Protocol Exchange, you are enabling researchers to more readily reproduce or adapt the methodology you use, as well as increasing the visibility of your protocols and papers. Upload your Protocols at www.nature.com/protocolexchange/. Further information can be found at www.nature.com/protocolexchange/about .

Please note that we encourage the authors to self-archive their manuscript (the accepted version before copy editing) in their institutional repository, and in their funders' archives, six months after publication. Nature Portfolio recognizes the efforts of funding bodies to increase access of the research they fund, and strongly encourages authors to participate in such efforts. For information about our editorial policy, including license agreement and author copyright, please visit www.nature.com/ni/about/ed_policies/index.html

Sincerely,

Ioana Visan, Ph.D.
Senior Editor
Nature Immunology

Tel: 212-726-9207
Fax: 212-696-9752
www.nature.com/ni